# Automatic Detection and Classification of Cardiovascular Disorders Using Phonocardiogram and Convolutional Vision Transformers

**DOI:** 10.3390/diagnostics12123109

**Published:** 2022-12-09

**Authors:** Qaisar Abbas, Ayyaz Hussain, Abdul Rauf Baig

**Affiliations:** 1College of Computer and Information Sciences, Imam Mohammad Ibn Saud Islamic University (IMSIU), Riyadh 11432, Saudi Arabia; 2Department of Computer Science, Quaid-i-Azam University, Islamabad 44000, Pakistan

**Keywords:** cardiovascular disease, heart valve disorders, continuous wavelets transform, vision transformers, deep learning, convolutional vision transformer

## Abstract

The major cause of death worldwide is due to cardiovascular disorders (CVDs). For a proper diagnosis of CVD disease, an inexpensive solution based on phonocardiogram (PCG) signals is proposed. (1) Background: Currently, a few deep learning (DL)-based CVD systems have been developed to recognize different stages of CVD. However, the accuracy of these systems is not up-to-the-mark, and the methods require high computational power and huge training datasets. (2) Methods: To address these issues, we developed a novel attention-based technique (CVT-Trans) on a convolutional vision transformer to recognize and categorize PCG signals into five classes. The continuous wavelet transform-based spectrogram (CWTS) strategy was used to extract representative features from PCG data. Following that, a new CVT-Trans architecture was created to categorize the CWTS signals into five groups. (3) Results: The dataset derived from our investigation indicated that the CVT-Trans system had an overall average accuracy ACC of 100%, SE of 99.00%, SP of 99.5%, and F1-score of 98%, based on 10-fold cross validation. (4) Conclusions: The CVD-Trans technique outperformed many state-of-the-art methods. The robustness of the constructed model was confirmed by 10-fold cross-validation. Cardiologists can use this CVT-Trans system to help patients with the diagnosis of heart valve problems.

## 1. Introduction

The main cause of death worldwide is cardiovascular disease (CVD), which claims more than 17 million lives each year [1]. CVD disease creates other pathological [2] issues with the heart, heart valves, or blood vessels. In this study, the authors describe a cost-effective and non-invasive technique for capturing heart signals through phonocardiography (PCG) [3,4]. It aids in enhancing the diagnosis of cardiac disorders and in creating new perceptions regarding the connection between the signal and the mechanical function of the heart. PCG signals can be used to diagnose a variety of CVD signals, including mitral stenosis (MS), mitral regurgitation (MR), aortic stenosis (AS), and mitral valve prolapse (MVP). A visual example of these PCG signals, categorized into five classes, is shown in Figure 1.

In practice, the visual screening of the PCG signal takes time [5] and is prone to error. Still, the arbitrary PCG signal inspection and analysis required by doctors requires substantial training and expertise. This encouraged the creation of a computer-aided diagnostic (CAD) method for the recognition of PCG signal-based cardiac screening and abnormality detection. CVD classification is currently a promising topic of research, based on biomedical signal processing and artificial intelligence (AI) [6]. Techniques utilizing AI can be utilized to get around these restrictions. Machine learning (ML) is a branch of AI that entails feature selection, statistical analysis, salient feature extraction (SFA), and classification. ML techniques are extensively used in combination with PCG signals to detect heart sounds [7]. Recently published papers for the diagnosis of cardiac illnesses used a variety of suggested research and methodologies [8,9,10]. Unfortunately, accuracy was not adequate, so the focus of attention shifted to developing a very accurate ML or DL for the diagnosing of cardiac problems. In the past, authors used a variety of feature extraction techniques and classifiers. However, these feature selection and classification techniques were hand-crafted, and frequently relied on iterative trial and error. To resolve this issue, deep learning (DL) techniques were developed.

Currently, DL algorithms are still used as the primary approach in detecting heart sounds, because smart detection PCG technology has not yet been widely adopted in actual clinical diagnosis. Therefore, advancements in the field of CVD diagnosis are facilitated by the study of, and deployment of, computer-aided (CAD) heartbeat detection techniques. In the past, cardiovascular disease was mostly detected using the following four steps: (1) preprocessing of the HS signals, (2) feature extraction, (3) feature selection, and (4) identification of normal and abnormal HS recordings.

It is difficult to categorize PCGs into five stages [11,12,13,14,15,16]. It is important to note that during the feature extraction step, several characteristics of one-dimensional signals are shared by various cardiovascular illnesses. The outcome of multi-classification may be impacted by these related properties. Therefore, it is crucial to emphasize the diversity of the various characteristics of heart disorders. There have been numerous manual feature extractions. Most of these hand-crafted features, such as amplitude, time interval, kurtosis, energy ratio, MFCC, entropy, etc., have physiological causes. Previous research often used these parameters to undertake binary categorization (normal PCG vs. abnormal PCG). This feature of manual computation is small and straightforward, but it might not be good enough for multi-classification and new databases.

As a result, there is a need to extract deep features for multi-class recognition. It is difficult to categorize five stages of CVD by using one-dimensional PCG signals. As a result, we employed a technique based on continuous wavelet transform-based spectrogram (CWTS) strategy to transfer energy from the PCG signal into 2D spectrogram images. Moreover, deep features were utilized in this paper to develop a classifier. To automatically extract more detailed information, several researchers have employed deep-learning models like CNN or other ANN models. In this study, we created a CNN model that is trained on discriminant representations of non-segmental PCG frames to offer a useful method for automatic detection. Our primary goal was to investigate alternate feature extraction methods for PCG classification, based on convolutional vision transformer (CVT), which is a combination of DL and vision transformer. The following are the novel contributions of this study:

We propose the convolutional vision transformer (CVT) architecture, based on local and global attention mechanisms, which is computationally efficient. It is possible to recognize 1-D signals in 2-D spectra for effective extraction of features.

A continuous wavelet transform-based spectrogram (CWTS) strategy is employed to transfer energy from the PCG signal into 2D images.

A simple and reliable attention-based transformer model is developed to classify multi-class PCG signals accurately and efficiently.

A classification of five designated categories is presented in this paper, based on heart sounds.

The rest of the paper is organized as follows. Section 2 introduces the details of previous work. Section 3 presents the proposed method, including PCG datasets, CWT, the transformation of signals into a spectrogram, feature extraction and selection, attention-based vision transformer model, and classification procedures. Section 4 presents experimental results. Section 5 and Section 6 provide some discussion and conclusions, respectively.

## 2. Literature Review

Heart sound segmentation (HSS), feature extraction (FE), and classification are the three phases that traditionally go into heart sound classification. The initial stage aims to locate the location of the basic heart sounds (HS). Each PCG recording is divided into several HD segments. The systolic and diastolic areas of the heart sounds are revealed by the precise localization of the HS. Segmentation is not required because the goal of abnormal HS detection is primarily to identify an abnormality in the heart sound, rather than to detect its presence. Therefore, a variety of strategies for classifying heart sounds without any segmentation have been suggested in the literature. When the segmentation information from the various strategies is used, they can attain equivalent results. A comparative performance of existing work for cardiac disease classification (CDC) is described in Table 1.

Regarding the second stage, numerous feature extraction algorithms have been proposed in the literature, falling into the following three primary categories: time domain [17], frequency domain [18], and time-frequency complexity domain [19]. Due to the physiological properties of the PCG signals, the time or frequency domain features are straightforward, simple to grasp, and easy to calculate. However, it can be challenging to quantify certain critical PCG signal information independently in the time or frequency domain. As a result, time–frequency (TF) domain feature extraction is growing in popularity. The TF-based features can offer more thorough information about the PCG signal and better feature extraction performance results, even though they require greater computing complexity than features based just on time or frequency [20]. Wavelet transformation, discrete and packet wavelet transform (DPWT), Hilbert transform (HT), empirical wavelet transform (EWT), variational mode decomposition (VMD), and adjustable Q-wavelet transform are some of the popular TF feature extraction techniques for PCG signals (TQWT). When the PCG signal’s TF matrix is generated using spine CT, it can more accurately capture pathological changes and offer superior resolution in the TF domain. However, due to the nonstationary and varied properties of PCG signals, such manually created features have their constraints, and feature extraction is still a difficult operation.

The final stage involves training a classifier on the retrieved characteristics to produce predictions for each PCG signal [21,22,23,24,25,26]. To categorize the HS based on extracted features, several machine learning-based classifiers have been proposed, such as the support vector machine (SVM), decision tree (DT), K-nearest neighbor (KNN), artificial neural network (ANN), multi=layer extreme learning machine (ML-ELM), hidden Markov model (HMM), etc. An ensemble of various classifiers was also used to further enhance classification performance. In [27], the authors suggested a tent-pooling decomposition and a graph-based feature generator to extract features. Five classes of PCG signals were classified using DT, linear discriminant, bagged tree, and SVM classifiers after iterative neighborhood component analysis (NCA) was used to determine the features. In [28], the authors chose the most discriminative features for NCA using a one-dimensional (1D) binary pattern with three kernels. For the classification of PCG signals, KNN and SVM were used. In [29], the authors took six audio variables from audio samples of PCG signals, including spectral centroid, zero crossing rate, energy entropy, spectral roll-off, volume, and spectral flux, and submitted them to four conventional machine learning-based classifiers for classification. Although PCG categorization has greatly improved thanks to machine learning-based techniques, these methods are still subjective and time-consuming [30]. Convolutional neural networks (CNNs) and long short-term memory (LSTM) are two deep learning models that have recently been used for the classification of heart sounds [30,31,32,33,34]. They have drawn more attention because of their automatic analysis and extraction of high-level representations from heart sounds. Additionally, it is becoming popular to identify PCG signals directly from entire audio recordings without first segmenting them.

**Table 1 diagnostics-12-03109-t001:** A Comparative performance of existing work for cardiac disease classification (CDC).

Cited Reference	* Dataset	Feature Extraction	Classification	Results	Limitations
Z.H. Wang et al. [17]	PRV	CWT + Spectrogram	LSTM-RNN	ACC: 93%	Five Classes
A.M. Alqudah [18]	PhysioNet and GitHub dataset	Instantaneous frequency-based features	RF and KNN	ACC: 95%	Five Classes
X. Cheng et al. [19]	Open Heart sound dataset	Heart sound segmentation features	Fisher ratio (FR). Finally, the Euclidean distance (ED) and the close principle	ACC: 96%	Two Classes
A. Rath et al. [20]	Pascal CHSE dataset	DWT and MFCC features	RF-MFO-XGB ensemble	ACC: 89%	Three Classes
J. Li et al. [21]	PRV	Multidimensional Scattering transform	PCA and Twin SVM	ACC: 98%	Two Classes
F. Khan et al. [22]	PhysioNet	Mel Frequency Cepstral Coefficients (MFCC)	ANN + LSTM	AUC: 91%	Two Classes
A.T. Saputra et al. [24]	-	PCA data correlation	NN and PSO	AUC: 98%	Two Classes
O. Arslan [25]	PRV	PWPT + EMD features	RF	ACC: 99%	Two Classes
J.S. Khan [26]	PhyioNet	Power Spectrum discriminating features	CNN	ACC: 98.89%	Two Classes
A. Yadav [29]	NIH	Spectral Statistical Features	SVM, k-NN, random forest, Naïve Byes	ACC: 97%	Two Classes
P. Dhar [34]	PhyioNet	Cross-wavelet transform (XWT)	Cross-wavelet transform (XWT) assisted Convolution neural network (CNN) utilizing the AlexNet model	ACC: 98%	Two classes

* PRV: Private dataset, DWT: Discrete wavelet transform, RF: Random Forest, CNN: Convolutional neural network, NN: neural network, PSO: Particle swarm optimization, SVM: Support vector machine, AUC: area under the curve, LSTM-RNN: long-term short-term memory recurrent neural network, CWT: continuous wavelets transform, PCA: principal component analysis.

Time, frequency, time–frequency (TF) features, energy features, and entropy features were all merged into feature vectors by the authors. For PCG classification, they were combined with the deep learning features that CNN had previously collected from pictures with Mel frequency cepstral coefficients (MFCCs). According to the authors, handcrafted characteristics could only reflect the differences in PCG signals brought on by HVDs from specific angles in practical applications. More thorough disease-related data could be gathered when deep learning characteristics, having good representation capabilities, were combined. In [35], the authors developed a new 2D CNN architecture for HS classification that could extract more discriminative features while using fewer parameters. This architecture included spatial and channel-wise attention methods. In addition, more recent studies have been reported to use PCG signals, spectrogram and deep learning techniques [36,37,38,39,40,41].

## 3. Materials and Methods

### 3.1. Acquisition of Dataset

To test and train the convolutional vision transformer (CVT), we used a dataset that is publicly available in the form of PCG signals. A total of 1000 PCG recordings from five different courses, each with 200 recordings, were produced. The various signal classes were N, AS, MS, MR, and MVP. Each recording was sampled at an 8000 Hz frequency.

This dataset was the phonocardiogram database [22]. The database has over 1000 audio recordings in wav audio format, though it is unknown how many individuals are included, and 8000 Hz is the sampling frequency. The other two types of heart sound signals (HSS) were normal (N) and four primary valvular heart disorders, including mitral stenosis (MS), mitral valve prolapse (MVP), mitral regurgitation (MR), and aortic stenosis (AS). In each category, there were 200 HD recordings (200 audio recordings). In database A, heart sound signals could last anywhere from 1.1556 to 3.9929 s. Based on the minimum time length of the HS signal in the dataset, we used the HS signal’s maximum time of 1.1556 s. The dataset of the five categories of original HS could be obtained in the repository: https://github.com/yaseen21khan/Classification-of-Heart-Sound-Signal-Using-Multiple-Features- (accessed on 10 September 2021).

All PCG recordings were sampled at an 8 kHz rate, and a subsequent de-noising preprocessing step was then applied. The noise that contaminates PCG signals typically comes from a variety of sources. Therefore, filtering noise to eliminate these distortions is crucial. This should be done so as to keep all diagnostic data needed for PCG signal processing, while deleting all unnecessary components known as noise. To reduce as much background noise as possible, the PCG signals were carefully filtered. Therefore, out-of-band noise was reduced by using a Butterworth bandpass filter with cut-off frequencies of 25 Hz and 900 Hz. A visual example of this five-stage PCG is displayed in Figure 2.

### 3.2. Proposed Methodology

This study investigated the classification of heart sounds into five distinct categories, including artifact, extra heart sound, extra systole, murmur, and normal, using a 2D convolutional neural network. With no segmentation of the heart sound waves, the goal of this research was to develop a trustworthy method for the automatic detection of heart valve diseases (HVDs). When using a deep learning (DL) model to detect anomalous patterns in PCG data, a CWT spectrogram was used to extract representative features. The immediate energy of the PCG signal was first divided into various sub-bands using CWT. These sub-bands, which served as discriminant traits, had preserved the oscillatory characteristics of PCG. Second, these features were converted from being one-dimensional (1D) to two-dimensional (2D) spectrograms using a continuous wavelet transform-based spectrogram (CWTS) strategy. Following that, a brand-new attention-based transformer architecture was created to categorize the spectrogram into five groups. To test the effectiveness of the suggested strategy, experiments using multi-class classification (normal vs. AS vs. MR vs. MS vs. MVP) were conducted using three freely available PCG datasets. Finally, various performance metrics were used to assess the categorization outcomes. The proposed algorithm’s flowchart is shown in Figure 2.

Research manuscripts reporting large datasets that were deposited in a publicly available database should specify where the data were deposited and provide the relevant accession numbers. If the accession numbers have not yet been obtained at the time of submission, it is necessary that they are provided during review. They must be provided prior to publication.

#### 3.2.1. Signal Preprocessing and Noise Removal

PCG signals frequently experience noise from sources such as lung noises, power frequency interference, electromagnetic interference from the environment, and interference from electrical signals with human body signals. The diagnosis of PCG recordings is made difficult, if not impossible, by these diverse noise components. Due to its features like multi-resolution and its windowing technique, continuous wavelet transform (CWT), which is a collection of high-pass and low-pass filters, exhibits exceptional performance in signal denoising.

The sample frequency of the dataset was 8000 Hz. However, the sampling frequency for database B was 2000 Hz. We did not pre-process database A. We preprocessed database B, keeping database A’s original signal intact. To reduce the difference in sampling frequency, the dataset sample frequency was reduced to 2000 Hz. After that, the sample length of the heart sound signals in the two databases was fixed at 2312. The heart sound recordings from database B contained a small amount of noise, but the signal quality in database A was excellent. As is well known, the frequency of a heartbeat typically ranges from 50 to 150 Hz [28]. Digital filters are used to filter out low and high-frequency components. In this work, a third-order Butterworth filter with a bandwidth of 15 Hz to 150 Hz was used to filter the HS signals. To remove noise beyond the bandwidth and avoid time delay, the filtered sequence was then reversed and put through the filter once again. The dataset’s signal was then normalized.

#### 3.2.2. Data Augmentation

The application of conventional data augmentation techniques to the field of heart sound signals (HSS) is hampered by the time series nature of the signal and the unique characteristics of each individual. Therefore, finding a more appropriate and efficient augmentation method than the original HSS was one of the main challenges in designing the multi-label heart sound (HS) diagnosis system. Data augmentation operations often involve flipping, rotating, reflecting, shifting, zooming, contrasting, coloring, and noise disruption [29,30,31]. However, data augmentation techniques in the realm of images only alter fundamental details, such as location and angle, from a macro-perspective and, therefore, are only applicable to straightforward computer vision techniques, like image recognition, and they cannot be used to augment data. The PCG augmentation approach used in this study utilized a 1D signal augmentation mechanism. In order to identify the model with a greater generalization performance, the augmentation approach incorporated HS signals in a variety of instances. The research investigated backdrop forms while also allowing transfer learning models to classify different heart sound signals even in noisy environments.

The heart sound signal “original signal” was a specific HSS. The equisized background transformations were produced stochastically at the same moment. “Delta” was used to represent a deformation control parameter, and “random signal” was used to represent a background transformation. The background deformation’s “delta” ranged from “(0, 1).” Equation (2) was used to calculate the augmented signal, which was produced by combining the original heart sound signal with random background noise. It should be emphasized that there was no data augmentation in the testing unit. The outcome of the augmentation of the data is shown in Figure 3. Figure 3a shows the initial HSS. Figure 3b shows the HSS that was enhanced using Equation (2), and Figure 3c shows the denoised signal of Figure 3b. Finally, there were 2400 PCG recordings altogether in the database. Each lesson contained 400 PCG recordings.

#### 3.2.3. Signal Transformation

To determine the time–frequency representation of a sound, time–frequency (TF) transformation is a popular method in the classification of speech events. Using TF representation, a one-dimensional (1D) signal is converted into a three-dimensional (3D) image. Following that, the most likely sound source is identified using the attributes that were derived from the transformation. Based on their analysis in [32], the authors drew the conclusion that the continuous wavelet transform-based spectrogram (CWTS) presented the TF content of PCG signals in the clearest representation [42]. Several authors’ analyses showed that the CWTS process and the signal were represented in the form of a spectrogram. The heart-sound signal’s magnitude spectrogram is computed for each sample. The transfer learning models were tested and trained using these spectrograms. We employed the CWTS process, compared to the technique in [17], for recovering heart sound signals, based on LSTM architecture.

The “scale” parameter of the wavelet transform could be altered to identify various frequencies in the signal, as well as their locations. We now knew the frequencies present in the time signal as well as their locations. Wavelets compressed at smaller scales could, therefore, collect higher frequencies. A wider scale, on the other hand, meant lower frequencies could be picked up. An illustration of a compressed and stretched wavelet can be seen in the image below. Superior time and frequency resolution was provided by the CWT. This enabled the use of various analysis windows of varying sizes and frequencies. The spectrograms of the heart sound signals showed the frequencies at different times and offered a visual representation that could be utilized to distinguish individual heart sounds. The CWT produced data for a spectrogram, and each RGB image was downsized to an array of size (n-by-m-by-3) to match the inputs of various deep learning (DL) algorithms. Figure 4 displays the six typical HS signal spectrograms. The spectrogram of the original heart sound signals (HSS) is shown in Figure 3’s time–frequency plot, and the spectrogram of the heart sound (HS) signals is shown in Figure 3 in the form of a color spectrogram.

The CWT, often referred to as the constant-Q transform, is a technique for time–frequency analysis that offers an equal-resolution time–frequency representation of a signal on a log–frequency scale (Figure 2). Particularly in the high-frequency range, the human auditory filter bank is known to have an equal resolution on a log–frequency scale as with the CWT. Hence, modeling, analyzing, and processing spectrograms obtained by the CWT would be one promising technique to enable computers to mimic the important functions of the human auditory system (a spectrogram). In fact, recent research has demonstrated that multiple fundamental frequency estimation performed exceptionally well in the magnitude CWT spectrogram domain.

On a logarithmic frequency scale, CWT spectrograms have an equivalent resolution to STFT spectrograms, but STFT spectrograms have the opposite. Compared to the short-time Fourier transform (STFT), the CWT has a much slower rate of calculation; this approach takes a very long time to complete. The reduction of computational complexity could be crucial in real-world circumstances. The phase might currently be quickly determined from a magnitude STFT spectrogram by the authors and collaborators [8]. The waveforms in the overlapping portion of succeeding frames must be constant when the hop size is less than the frame length. By implication, an STFT spectrogram is a redundant representation. Therefore, for the STFT spectrogram to be connected to a time-domain signal, a specific requirement must be met. This requirement has been referred to as the consistency requirement. In [8], we demonstrated how the issue of determining the phase from a magnitude STFT spectrogram might be framed as a conundrum of consistency criterion optimization. It specifies the deviation of any complicated array from this requirement. It soon became apparent that the algorithm was the same as the well-known technique put forth. We were able to introduce a quick approximation technique and provide a very clear demonstration of the algorithm’s convergence thanks to the formulation derived from the idea of spectrogram consistency. We might be able to make the most of the spectrogram consistency notion to create a quick approximation for phase estimation from a magnitude CWT spectrogram, because a CWT spectrogram is also a redundant representation of a signal.

The ECG signal’s spectrum density is described by a spectrogram over time. It displays a time–frequency domain signal. It is determined mathematically by the squared magnitude of the signal’s short-time Fourier transform (STFT), as calculated by Equation (1):(1)$$Spect\left(t,w\right)={\left|STFT\left(t,w\right)\right|}^{2}$$
where, the function $Spect\left(.\right)$ is the defined spectrogram over time, *w* parameter stands for frequency (in radians/seconds) and *t* stands for time (in seconds). A signal’s local segments’ sinusoidal frequency and phase content are estimated by STFT as a time-varying function. Long signals are divided into smaller chunks, and each segment’s Fourier transform is computed. As a result, during a limited amount of time, a spectrogram represents the time–frequency intensity spectrum.

#### 3.2.4. Features Extraction and Classification Using CVT

After successful completion of preprocessing, signal transformation using continuous wavelet transform-based spectrogram (CWTS) was performed, and the classification was performed by using a convolutional vision transformer (CVT) algorithm to recognize CVD into the following five classes of PCG signals: aortic stenosis (AS), mitral regurgitation (MR), mitral stenosis (MS), mitral valve prolapse (MVP), and normal. To develop the CVT model, we employed Inception v3 as a pretrain model by removing the last layer, as that layer is used to extract deep features. Compared to other pre-trained models, such as VGG16 or VGG19, this paper used the Inception v3 architecture to influence computational efficiency and ensure low parameters. The steps for classification are explained in the following paragraphs.

The Vision Transformer (ViT) [43] is probably the first entirely transformer-based design for vision, treating image patches as simple word sequences that are then encoded using a transformer. The ViT can produce impressive results in image recognition when it is pretrained on huge datasets. However, ViT has been found to perform poorly in image recognition without significant pre-training. This is a consequence of the Transformer’s strong model capability and lack of inductive bias, which causes overfitting. In multiple subsequent studies, sparse Transformer models designed for visual tasks like local attention have been investigated to effectively regularize the model’s capacity and enhance its scalability. One such effective attempt to change transformers by applying self-attention to shifted, non-overlapping, windows is the Swin Transformer. For the first time, with a pure vision transformer, this methodology outperformed ConvNets on the ImageNet benchmark. Window-based attention was found to have limited model capacity, due to the loss of non-locality, and, thus, scales negatively on bigger data regimes, like ImageNet-21K, despite being more adaptable and generalizable than the complete attention utilized in ViT. However, full-attention acquisition of global interactions in a hierarchical network at early- or high-resolution stages involves computationally intensive effort, since the attention operator has quadratic complexity. It is still challenging to include global and local interactions to balance model capacity and generalizability within a computing budget.

Shift, scale, and distortion invariance are characteristics of convolutional neural networks (CNNs) that were transferred to the ViT architecture [44], while retaining the benefits of Transformers (i.e., dynamic attention, global context, and better generalization). Even though vision transformers are successful on a large scale, their performance is still inferior to that of smaller CNN competitors (such as ResNets) when trained on less input. One rationale might be that CNNs are better suited to addressing vision-related issues because they naturally possess certain desirable qualities that ViT lacks. By utilizing local receptive fields, shared weights, and spatial subsampling, a texture compels the capture of this local structure and, as a result, also achieves some degree of shift, scale, and distortion invariance. Images, for instance, often contain a strong 2D local structure with highly connected spatially neighboring pixels. Additionally, the hierarchical structure of convolutional kernels enables the learning of a variety of complex visual patterns, from low-level edges and textures to higher-order semantic patterns that incorporate local spatial context.

Convolutional Transformer Block contains the convolutional projection as the first layer. In this study, we proposed that convolutions could be strategically added to the ViT structure to improve performance and robustness, while maintaining a high level of computational and memory efficiency. As proof of our hypothesis, we proposed our convolutional vision transformer (CvT), which integrated convolutions into the transformer and was intrinsically efficient in terms of parameters and floating-point operations (FLOPs). Compared to CvT in [44], we integrated an attentional selective fusion (ATTSF) layer to provide more focus on local and global interactions of pixels. In the original CvT model, the authors used a complex strategy by integrating token embedding and projection for Attention hierarchical transformers. However, we used a computationally efficient approach through an ATTSF mechanism.

The proposed convolutional vision transformer (CVT) model consists of two components: VT and feature learning (FLs). The FL extracts learnable features from the continuous wavelet transform-based spectrogram (CWTS). The learning features are fed into the VT, and, for the final detection stage, the VT transforms them into a series of picture pixels. The feature learning (FL) component is a collection of convolutional operations. The FL component follows the hierarchy of the Inception v3 architecture. The FL component differs from the Inceptionv3 model in that it does not have the fully-connected layer (FCL) seen in the Inceptionv3 architecture, and instead serves the purpose of extracting CWTS features for the VT component, rather than classification. The result is a CNN without the FCL layer. In addition, we integrated an attentional selective fusion (ATTSF) consisting of global attention and local attention, which could add more flexibility when fusing various forms of information. In this study, we first fused the LBP features (LBP) and the CNN features (CNN) to get more local and global interactions.

With a kernel size of 3 × 3, the feature learning (FLs) component contained 17 convolutional layers. The CWTS images’ low-level features were extracted using the CNN layers. A step and padding of 1 were used in all CNN layers. All of the layers employed the ReLU activation function for non-linearity and batch normalization (BN) to normalize the output features. The BN function normalized changes in the distribution of earlier layers because these changes had an impact on how the CNN architecture learned. Additionally, a 2-by-2-pixel window with a stride of 2 was pooled up to five times. The max-pooling process cut the image’s dimension in half. With the first layer, the convolutional layer’s (channel’s) width was doubled by a factor of two after each max-pooling operation. The VT component of this CVT model received a feature map of the CWTS spectrogram as input. Seven patches were created from the feature maps, which were then embedded into a linear series of lengths of 1 × 1024. The positional information of the image feature maps was then retained by adding the embedded patches to the position embedding. The position embedding had a dimension of 2 × 1024. The position embedding and patch embedding were received by the VT component, which then sent them to the transformer.

In contrast to the original Transformer, the Vison Transformer merely made use of an encoder. The MSA and MLP blocks made up the VT encoder. The block of MLP was an FFN. The transformer’s internal layer was normalized by the Norm. The Transformer had eight heads of attention. The ReLU nonlinearity and two linear layers made up the MLP head. A typical CNN architecture’s fully linked layer and the MLP head task were equivalent. A total of 2048 channels were present in the first layer, and two channels were present in the last layer, which represented the class of cardiovascular diseases. The CVT model comprised 38.6 million learnable parameters and a total of 20 weighted layers. For the final detection goal, Softmax was used on the MLP head output to compress the weight values between 0 and 1.

An overview of the convolutional visual transformers can be broken down into three sections: classification of expressions, relationship modeling, and visual word extraction. The foundation for feature map extraction was pre-trained by using Inception v3. Our Attentional Selective Fusion fused all the extracted features to produce representative visual words. Simply flattening the feature map’s spatial dimensions and projecting to the desired dimension yielded the input visual words. Additionally, we used a multi-layer transformer encoder to model the connections between various visual feature components. Finally, using a simple softmax function, the network predicted expression.

As shown in Figure 4, our attentional selective fusion (ATTSF) consisted of global attention and local attention, which could add more flexibility when fusing various forms of information. As stated, we first fused the LBP features (LBP) and the CNN features (CNN), deriving two feature maps (LBP, CNN) retrieved from the backbones, to capture the next information interaction:(2)$$Features\_map\;\left(i,j\right)=WL\times LBP+WC\times Cnn$$
where *Features_map* follows the addition of the *LBP* and *Cnn* is the integrated feature maps, and + denotes element-wise addition. The weights for the initial integration, *WL* and *WC*, were easily produced by two 1 by 1 convolutions. Then, to do both global and local selective fusion, we chose global average pooling and pixel-wise convolution as the global context and local context aggregators, respectively. The global context gradually converted each feature map of size (*H* × *W*) into a scalar, making use of the feature inter-channel interactions. Given that it preserved and highlighted the input pieces’ subtler distinctions, the local context was a useful addition to the global context. Aggregating local and global contexts might enable the network to take advantage of several types of information and more precisely recognize ambiguous signals. Both the global and local contexts were calculated.

## 4. Results

PCG signals frequently experience noise from sources, such as lung noises, power frequency interference, electromagnetic interference from the environment, and interference from electrical signals with human body signals. The diagnosis of PCG recordings is made difficult, if not impossible, by these diverse noise components. To remove such noises, we employed a multi-resolution and windowing technique, by using the continuous wavelet transform (CWT) along with a collection of high-pass and low-pass filters, which exhibited exceptional performance in signal denoising. To test and train the proposed system, we used 2400 PCG sound signals, which were doubled by using data augmentation techniques. Afterwards, the time–frequency content of PCG signals was represented most clearly among the three time–frequency representations (short-time Fourier transform (STFT)) to transfer 1D signals into 2D spectrograms. We applied the characteristics of convolutional neural networks (CNNs) to the Vit architecture [44] to recognize five classes of CVD diseases, while retaining the benefits of transformers. A machine with 32 GB of RAM and an Intel Core i7 6700K processor was used for the experiments. The Keras platform and TensorFlow backend, powered by GPUs, were used by the suggested deep learning model.

### 4.1. Performance Evalaution

Metrics were used to measure the proposed solutions and the existing solutions. The first measurement was *Accuracy* (ACC). This was the metric used to evaluate the classification models, a way of measuring how well the algorithm classified the data. Optimizing the model’s accuracy lowered the cost of errors, which can be huge. In addition, it could be calculated as follows:(3)$$Accuracy=\;\left(TP+TN\right)/\left(TP+TN+FP+FN\right)\times 100$$

True Positive (*TP*) and True Negative (*TN*) showed that the algorithm correctly predicted the data, either true or false. As for false, a False Positive (*FP*) and False Negative (*FN*) showed that the algorithm predicted the data incorrectly, whether true or false. The second measurement was Sensitivity (SE), which was defined as the percent of actual positive instances that were virtually predicted as positive, indicating that there was a percent of actual positive instances that were misclassified as negative. It is worth noting that this measurement implied that a low *FN* rate almost always accompanied a high recall. Sensitivity is also known as Recall, True Positive Rate and Hit Rate. The following formula can be used to compute it:(4)$$Recall=TP/\left(TP+FN\right)\times 100$$

The third measurement was *Specificity* (SP), defined as the percent of true negative to total negative in the data. The formula for this measurement is as follows:(5)$$Specificity=TN/\left(TN+FP\right)\times 100$$

The fourth measurement was *Precision*, which refers to the ability to correctly identify positive categories among entire expected positive classes, as measured by the proportion of all correctly predicted positive categories to all accurately expected positive categories. Mathematically, it can be represented as follows:(6)$$Precision=TP/\left(TP+FP\right)\times 100$$

The fifth measurement was the *F1*-*Score*. This metric is important for determining the exactness and robustness of a classifier. The *F1*-*Score* is a critical metric for performance assessment that considers both recall and precision. It can be calculated as follows:(7)$$F1-Score=2\times \left(precision\times recall\right)/\left(precision+recall\right)$$

### 4.2. CVT Model Training

The huge PCG signal data were used for training, validation, and testing of the proposed CVT-Cardio system. Validation was the foundation for both training and testing. In this study, the CVT models were trained using nine folds and then tested using one-fold. To guarantee that the whole dataset was covered for training and testing situations, the process was repeatedly iterated. Since all the patient recordings were combined into one collection, the selection of each folder was not based on distinct subjects. The training data included 90% of the original heart sound data and all the augmented data. There were 3800 heart sound recordings, 2000 of which were enhanced (augmented), and 1800 of which were the original recordings. By experimental analysis, we selected 2000 recordings to perform training, testing and validation.

### 4.3. Fine-Tunning

To enhance the model, we used the vision transformers’ [30] fine-tuning approach. To fine-tune, an SGD optimizer with a momentum of 0.9 was employed. We pre-trained our models at a resolution of 224 × 224 and fine-tuned at a resolution of 384 × 384, like vision transformers. With a total batch size of 512, we refined each model individually over 20,000 steps on the PCG dataset.

### 4.4. Computational Analysis

To measure the performance of the proposed CVT-Trans system, we computed the running time. To show that the suggested technique was a computationally efficient model, it was developed and trained on a GPU-based system, rather than a CPU-based system. For all datasets, the durations for STFT calculation, CVT with time domain input, and CVT-Trans with frequency input were computed. On average, this step took 0.4 Ms. Overall, a CWT spectrogram transformed steps from original 1-D PCG signals taking, on average, 1.2 MS. This point-of-view is visually explained in Figure 5a. An attention-based CVT transformer architecture was created in this study to categorize the spectrogram into five groups. On average, this step took 1.2 MS. The key benefits of the proposed technique were quick classification and STFT computation, excellent accuracy acquired by utilizing all datasets, and a minimal number of layers. The original TL models contain more parameters and FLOPs (given in Figure 5b), compared to the suggested CVT-Trans model.

The outcome was a compact model that required fewer computational support systems for implementation. Training time complexity = O (n × m × d), where, the parameter n is the input dimension, d is the batch size, and m is the output dimension. In general, the proposed CVT-Trans model took linear running time calculated as O (n × m × d). By utilizing tensor processing units (TPUs), which were offered by the Google cloud, this time complexity could be further decreased. In actual use, the TPUs significantly increased DL model speeds while using less power. This viewpoint will also be covered in further studies.

### 4.5. Results Analysis

The authors’ approaches only considered a single representation of heart sound signals, a spectrogram from the preprocessed heart signals that were captured through PCG signals. The time–frequency domain image of a sound signal is called a spectrogram. The inputted heart sound waves were transformed into the appropriate spectrogram before being further categorized, using transfer learning models, into five categories (Table 2). Table 3 lists the various approaches that do not include transfer learning. The various classifications of cardiac disorders demonstrate the accuracy of alternative methods in comparison to transfer learning models. On the same database, these various methodologies were compared against one another. However, the outcome of Table 4 demonstrates that the accuracy was up-to-the-mark.

Figure 6 displays the accuracy with loss of the proposed training and testing in the enriched-PCG and the original-PCG databases, respectively, for one transfer learning model. The 10-fold cross-validation method was used to conduct the experiment. Table 3 shows these measures for each fold, including training and testing samples: training accuracy (Acc), testing accuracy (Val Acc), training loss (Loss), and testing loss (Val Loss). The results of 10-fold cross-validation on the expanded-PCG database using data augmentation are shown in Table 3. The results of 10-fold cross-validation on the original database are shown in Table 4, using two categories. The categorization of five classes using the proposed approach, both with supplemented data training and with original data training, achieved an average accuracy of 98%, as shown in Table 3 and Table 4. The PCG database’s results demonstrated the effectiveness of this approach. We additionally assessed the effects of the PCG augmentation technique using additional background deformation.

A machine with 32 GB of RAM and an Intel Core i7 6700K processor was used for the experiments. The Keras platform and TensorFlow backend, powered by GPUs, were used by the suggested deep learning model. The efficacy of the suggested strategy was tested in a few trials. The 10-fold cross-validation technique, which reduced the variance of the estimate for the classifiers, was used to evaluate the classification outcomes. The training and testing portions of the fragmented data were separated. The data set was split into ten subsets for the 10-fold cross-validation, making sure that each subset contained the same number of observations with a certain categorical value. The other night subgroups were combined to create a training set each time, with one of the 10 subsets being used as the test set. As a result, each fold was utilized ten times for training purposes and once for testing purposes. The average of the 10 implementations was the outcome. It confirmed whether the suggested method outperformed others. Table 2 demonstrates the results obtained by the proposed CVT-Trans system to classify PCG signals into five classes.

Different strategies were employed in the past to classify PCG signals into AS, MR, MS, MVP, and N categories by using machine learning, feature engineering, and deep learning techniques. However, a thorough investigation of the efficacy of classification across huge datasets was not considered. Therefore, it was more desirable to determine if the suggested technique could classify PCG signals that contained various combinations of datasets. Seven distinct classification examples were offered from the datasets to handle this issue. Table 4 tests all centers on identifying normal and abnormal PCG signals by using a 10-fold cross-validation test. We achieved the best classification results for two classes of normal and abnormal PCG signals by using data augmentation techniques.

Figure 7 and Figure 8 show the categorization outcomes for five cases using a 10-fold cross-validation methodology. The classification results obtained by the classifier for binary and multi-class classification problems were thoroughly distributed in a confusion matrix. The 10-fold cross-validation’s overall confusion matrices are illustrated in Figure 7 and Figure 8. As shown in Table 4 and Table 5, our study showed that accuracy increased when distinguishing between binary and multi-class PCG signals. Overall classification accuracy for the experiment’s five binary classification cases was 100%, 99.00%, 99.75%, 99.75%, and 99.60%, respectively. The robustness of our model was verified and had the benefit of discriminating features. In addition, the features were automatically retrieved through the CVT-Trans model, which provided a low misclassification rate.

Comparisons were also performed using state-of-the-art approaches, such as those of Wang RNN et al. [17] and Cheng RF et al. [19]. The authors [17] used LSTM and RNN deep learning models with the CWT-spectrogram technique to recognize cardiovascular disease categories. Whereas in [19] the authors used different heart sound features to recognize two classes of PCG signals, we selected [17,19] because they were easy to implement, compared to other systems. Figure 8 demonstrates the results obtained by these two techniques compared to the proposed CVT-Trans system. The proposed CVT-Trans system outperformed both techniques.

Additionally, for both the four-class and five-class classifications, we were able to obtain a total classification accuracy of 98.87% and 98.10%, respectively. In comparison to cutting-edge methods, the proposed (PRS) system could effectively distinguish between various classes of PCG signals. In another experiment, we compared recent studies, such as those of Z.H. Wang et al. [17], and A.M. Alqudah [18], as shown in Table 5. The obtained result demonstrated that our CVT-Trans outperformed the rest.

Several studies with comparable data sizes to ours have achieved results that were less accurate than ours. Our model had slightly better accuracy, sensitivity, and specificity values. Moreover, the past studies developed mostly two-classes of PCG signals by using hand-crafted features. We employed five-classes of heart sound classification, and the features were not manually extracted. In our work discriminating characteristics were automatically picked up by the model, rather than the more laborious process of retrieving and selecting features manually. Additionally, our model classified each sample in less than a millisecond. Our approach was, thus, quicker and more suited for quick diagnosis. Our large data size also required 10-fold cross-validation, whereas 5-fold cross-validation was employed in previous studies. The size difference between the training set and resampling subsets became smaller with a bigger k value of 10, which lowered the bias of our suggested method. This demonstrated that our proposed strategy was more dependable. The confusion matrix revealed that each class’s misclassification rates were less than 3%. This low misclassification rate attested to the model’s great accuracy.

The suggested CWT spectral loss may be used to train a high-quality model, according to experimental data. It is difficult to balance the time and frequency (TF) resolutions in STFT, since they depend on the frameshift and analysis window design. The CWT may be used to do time–frequency (TF) analysis at various temporal and frequency resolutions. It is also feasible to modify the CWT’s time–frequency analysis in order to take into account scales resembling those used in human vision. In this study, employing both STFT and CWT spectral losses, we examined the training of high-quality features from waveform models. CWT can consider temporal and frequency resolutions that are closer to human perception scales than in STFT. The nonlinear nature of PCG signals cannot be adequately analyzed by the methods now in use in any of these fields. This study, in which the continuous wavelet transform (CWT), and STFT spectrogram methods were employed together to analyze nonlinear and non-stationary aspects of the PCG signals, overcomes these restrictions. To find time–frequency maps, two prominent techniques were used: the short-time Fourier transform (STFT) and the continuous wavelet transform (CWT). This illustration demonstrated how the CWT analyzed signals concurrently in time and frequency, as seen in Figure 9. The example demonstrated how the CWT worked better than the short-time Fourier transform (STFT) when localizing transients. The example also demonstrated how to approximate time–frequency localized signals using the inverse CWT. The phase information of the signal under analysis was not provided by the CWT. In this paper, we used the CWT and STFT algorithms with the intention of comparing them to earlier STFT findings. The scale factor and the mother wavelet selection served as adjustment parameters. When the settings were changed, a combination of CWT and STFT was used to automatically diagnose CVD disease. Finally, a comparison between STFT and CWT was suggested by considering the precision of detection and time processing. Figure 9 illustrates a time–frequency map made possible by CWT and STFT spectrograms.

Figure 10 shows the comparisons of different CNN-based pretrained models with the proposed CVT-Trans technique for classification of five categories of cardiovascular disease. Figure 10a shows the testing/validation/training loss, and Figure 10b represents the performance measures with computation time. This figure shows the efficiency of the proposed CVT-Trans model. The relationship between the true positive rate (TPR), which is known as sensitivity, and the false positive rate (FPR), which is known as 100% specificity for various classifications, is explained by the receiver operating characteristic (ROC) curve. The total area under the ROC curve (AUC) can be used to assess how well the categorization performed. In this work, the AUC was determined for each iteration and provided as a complete quantitative evaluation of classification performance using 10-fold cross-validation. The ROC curves and associated AUC for all seven instances are shown in Figure 11 to better clarify the 1D CNN classifier’s performance. Overall, our classification strategy performed well. The suggested pattern recognition system (PRS) could successfully distinguish between various PCG signal classes using CWT, spectrogram, and CVT techniques.

We also performed experiments to show the importance of the proposed CVT-Trans system compared to state-of-the-art approaches, such as Cheng-FRED [19], Rath-RF-MFO-XGB [20], Li-PCA-TSVM [21], Khan-ANN-LSTM [22], Saputra-NN-PSO [24], and Arsalan-RF [25], in terms of SE, SP, F1-score, RL, PR, and ACC measures. The standard hyper-parameters were defined, as presented in the corresponding studies. Firstly, the comparisons were performed between the proposed method and state-of-the-art techniques by using data augmentation and a 10-fold cross-validation test, in terms of two stages of CVD disease, as shown in Table 6. As displayed in Table 6, the Khan-ANN-LSTM [22] system obtained good classification results (SE of 0.88, SP of 0.87, F1-score of 0.88, RL of 85%, PR of 88%, and ACC of 86%) for CVD heart disease, compared to the other approaches. However, the proposed CVT-Trans method achieved 100% ACC. Secondly, we also measured the performance of the CVT-Trans system in terms of the recognition of the five stages of CVD heart disease. The results are mentioned in Table 7. As mentioned in Table 7, the proposed CVT-Trans system outperformed (SE of 0.99, SP of 0.98, F1-score of 0.99, RL of 98%, PR of 98%, and ACC of 100%) the state-of-the-art approaches. The good results obtained were because we developed a convolutional vision transformer (CVT) architecture based on local and global attention mechanisms in a continuous wavelet transform-based spectrogram (CWTS) strategy. The developed strategy was effective and efficient compared to many state-of-the-art systems.

## 5. Discussion

When one or more of the four heart valves are damaged or flawed, it is called heart valve disease (HVD). The pulmonary, aortic, mitral, and tricuspid valves are the four valves of the human heart [20]. The heart’s mechanical activity and function are good, and blood backflow is prevented when the heart valves open and seal correctly. HVD can result from any type of heart valve damage. Aortic stenosis (AS), mitral stenosis (MS), mitral regurgitation (MR), and mitral valve prolapse are examples of common HVDs. 

A full comparison with some excellent recent work is included in Table 1. There are significant constraints around HSs classification because there are so few clinical datasets. Additionally, much research has focused on binary categorization. Additionally, a single database, such as the Open-Heart Sound, was used for most validation and training. There are no databases with categories of heart sounds that have various labels for the sounds. The database from the website and the data we acquired ourselves were combined to address this issue. Five different types of heart sound signal were included in this database: normal, mitral stenosis, mitral regurgitation, mitral valve prolapse, and aortic stenosis. Additionally, the proposed methodology was verified in our study using a data augmentation scenario. This made excellent use of numerous noise recordings and the heart sound amplification approach.

Compared to other modalities, phonocardiography (PCG) is non-invasive, non-destructive, repeatable, easy to use, and inexpensive. It can be used for early detection, long-term intensive care, and prevention of diseases associated with the heart. The growth of digital medical technology and biological technology has increased the demands on the processing and analysis of heart sound data in key sectors. Automatic analytic techniques for processing medical sequence signals might share the burden and pressure of the medical sector, and these systems can also provide long-term disease monitoring. They can also assist medical professionals in better understanding the situation and in developing measures for illness avoidance and treatment. Cardiologists can improve society’s general health in this way.

The automatic diagnosis of heart sounds has come a long way, but there are still several obstacles to be addressed before the technology can be fully developed. For instance, there are many heart disease classifications that are inaccurate and have large feature extraction and database limitations. By overcoming these obstacles, deep learning technology may make a significant advancement in the realm of human health. Our study provides the first pulmonary hypertension-specific heart sound database, often known as a “heart sound database of pulmonary hypertension.” Another drawback is that it sometimes takes a long time to gain heart-sound feature extraction. In order to automatically extract features using a convolution layer in the heart sound domain, we also proposed 1-D signal transfer to the 2-D image for training and testing. Finally, we suggested using convolutional vision transfer (CVT) techniques to accurately identify various heart sounds. By using prior information to address related tasks, this breaks the independent learning pattern. Transfer learning is crucial in the artificial intelligence field when dealing with tiny sample sizes of data since pre-trained weights might be more effective in training and achieve greater performance. In this study, we hypothesized that the diversity of the enriched data may help the networks generalize to previously unexplored material during the training phase. The robustness of training can be increased through data augmentation. Transfer learning and conventional approaches were contrasted. According to numerous performance indicators across all experiments, transfer learning networks outperformed other basic networks, such as CNN and the LSTM network. Physicians may be given an effective and precise way to triage patients using this strategy.

The suggested method significantly influences clinical settings by supporting doctors in making decisions regarding various cardiac diseases. Our model is effective at predicting when an anomaly appears in a recorded signal. Additionally, it was evaluated in regard to people with certain valvular disorders where early diagnosis is difficult. In conclusion, this paper has the following three contributions: (1) A novel way to transfer heart-sound signals into 2D spectrograms after preprocessing. Additionally, our approach was tested by using data augmentation scenarios. (2) An approach to augmentation enhancing the accuracy of the detection of cardiac disease, particularly in loud situations. (3) CVT learning, which is hardly ever used in the field of classifying heart sounds, according to the published literature. To validate the categorization strategies, we compared them with state-of-the-art models. For multiple categorizations of heart illnesses, we achieved a low mistake rate and high accuracy (0.98 accuracy for six categories of heart sounds), which helped to address multiple classification concerns. With an overall accuracy of 96.25%, a multi-class composite classifier was created using class-specific closest neighbor distance and class-specific residuals. Compared to other approaches, ours had a 98.87% overall accuracy rate.

The following crucial points are just a few of the numerous elements that contributed to the proposed method’s successful categorization performance. The CVT-Cardio could improve the performance of the transient characteristics, due to its good and instantaneous signal adaptation and high time resolution. The adaptable CWT could be used for a variety of applications, including the detection of HVDs, because of its variable and scalable time–frequency distribution properties. Based on the derived discriminant features, a trustworthy 2D CNN model for categorizing various PCG recordings was created. It has a straightforward structure and minimal computational complexity.

Other approaches provided higher classification accuracy (above 98%). However, they only conducted two-class research. The accuracy was 97%, which is similar to ours. Other research in Table 1 also found lower accuracy for two-class studies. The classification results from three-class investigations are summarized in Table 3. All the papers reported classification accuracy levels that were lower than ours. The other authors employed a comparable amount of data to attain the accuracy shown in Table 4, which was identical to ours, despite our model’s somewhat lower sensitivity and specificity values. Characteristics from PCG signals were first extracted using conventional machine learning methods, like the discrete wavelet transform, before being sent to the deep neural network in these other works. In contrast to our work, where discriminatory characteristics were automatically selected by the model, this would be more laborious because the features would need to be retrieved and selected manually. Additionally, our model classified each sample in less than a millisecond. Our approach is, therefore, quicker and more suited for quick diagnosis. Additionally, the 10-fold cross-validation for our enormous data size changed to 5-fold validation. The size gap between the training set and the resampling subsets decreased with a bigger k value of 10, which reduced the bias of our proposed method. This proves that our suggested strategy is more reliable. The confusion matrix revealed that each class’s misclassification rates were less than 10%. This low rate of misclassification attests to the model’s great accuracy.

The highest accuracy, 99%, was attained by the proposed CVT-Trans with a pre-trained Xception net. As a result, it is appropriate to create AI-based machinery for the early detection of cardiovascular diseases. The research’s novelty lies in the use of the CVT-Trans network model for valvular heart sound analysis, which requires very little time in training and testing. However, several uses in research papers pertaining to video/image analysis had previously been discovered.

## 6. Conclusions

Heart valve disease is becoming more common, and it has a higher fatality rate than other cardiovascular disorders. Heart valve disease patients’ PCG signals contain crucial information that can be used for diagnosis. A novel patch-embedding technique (CVT-Trans) based on convolutional vision transformer divides the PCG signals into five classes, such as normal (NRM), aortic stenosis (AS), mitral valve prolapse (MVP), mitral stenosis (MS), and mitral regurgitation (MR). Continuous wavelet transform (CWT) and spectrogram are used to extract representative features from PCG data. The instantaneous energy of the PCG signal is first divided into various sub-bands using the CWT. To develop the CVT-Trans model, we first converted preprocessed 1D signals into 2D through an STFT spectrogram approach. Following that, an attention-based transformer architecture was created to categorize the spectrogram into five groups. We used PCG sound recordings to analyze and assess the CVD-Trans system. The five-class classification had an overall average accuracy ACC of 100%, SE of 99.00%, SP of 99%, and F1-score of 98%. The signals in our study were utilized to train a deep model for the categorization of several heart valve disease types, including normal, MVP, MS, MR, and AS. In the classification of “normal,” a high training accuracy of 97% was attained, which yielded the greatest accuracy of 98.20%. The model’s robustness was confirmed using 10-fold cross-validation. One of the first experiments to classify heart sounds was based on five categories. Cardiologists may be able to utilize the suggested model as a diagnostic tool to find heart valve disease. A PCG signal’s problematic status can be accurately determined using a CNN-based deep learning algorithm. It proved to be the best method for classifying the various medical disorders affecting the heart. The efficiency of the suggested method in differentiating between normal and multi-class diseased heart activity, in comparison to other state-of-the-art methods, was demonstrated by extensive trials on a real-world PCG database.

Future studies will expand on this technique to use PCG signals to identify stress and coronary artery disease. In addition, the effect of SARS-CoV19 on cardiovascular disorders will also be analyzed using the CVD-Trans technique. The CVD-Trans system can potentially be used to detect HVDs in real-time for smart healthcare and Internet of Medical Things (IoMT) applications.

## Figures and Tables

**Figure 1 diagnostics-12-03109-f001:**
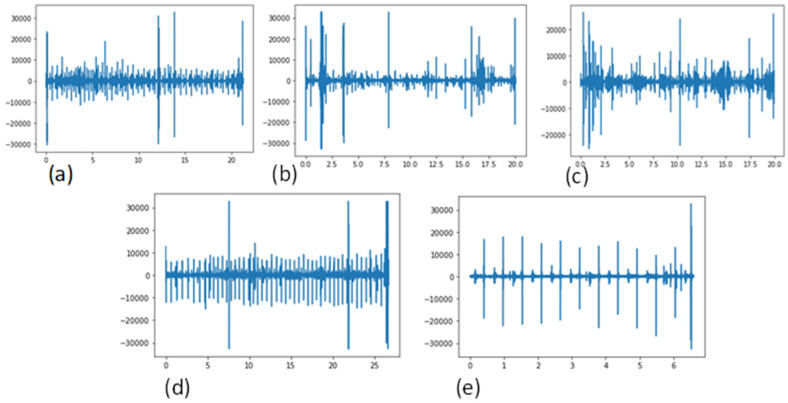
Signals of the existing CVD classes, using a phonogram, (**a**) Aortic stenosis (AS), (**b**) Mitral regurgitation (MR), (**c**) Mitral stenosis (MS), (**d**) Mitral valve prolapse (MVP) and (**e**) Normal.

**Figure 2 diagnostics-12-03109-f002:**
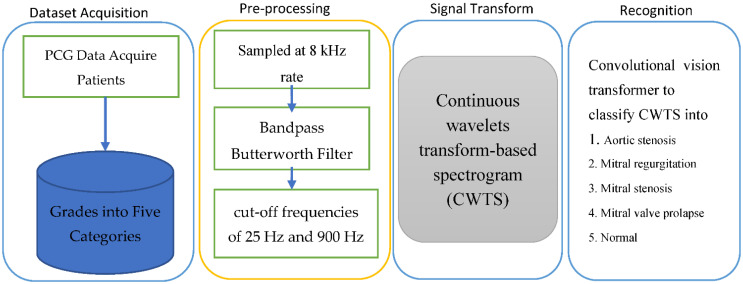
A systematic flow diagram of proposed CVT-Trans system for classification of the CVD classes.

**Figure 3 diagnostics-12-03109-f003:**
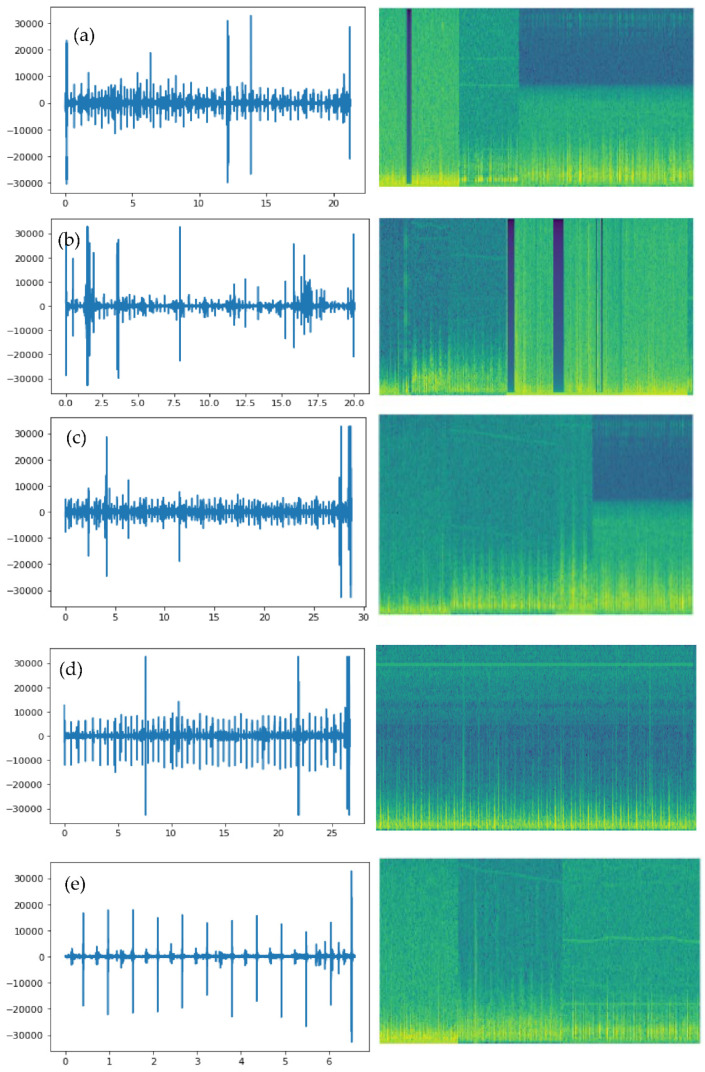
An example of a signal transform step after preprocessing through the continuous wavelets transform-based spectrogram (CWTS) approach. (**a**) Aortic stenosis (AS), (**b**) Mitral regurgitation (MR), (**c**) Mitral stenosis (MS), (**d**) Mitral valve prolapse (MVP) and (**e**) Normal.

**Figure 4 diagnostics-12-03109-f004:**
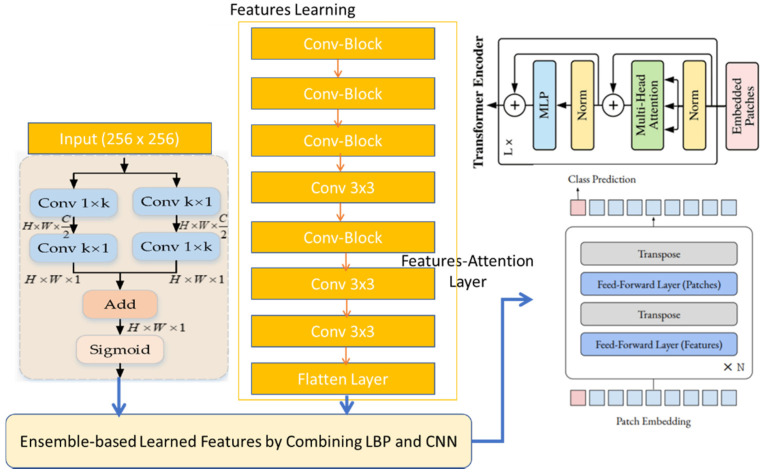
The pipeline of the proposed convolutional vision transformer (CVT) architecture.

**Figure 5 diagnostics-12-03109-f005:**
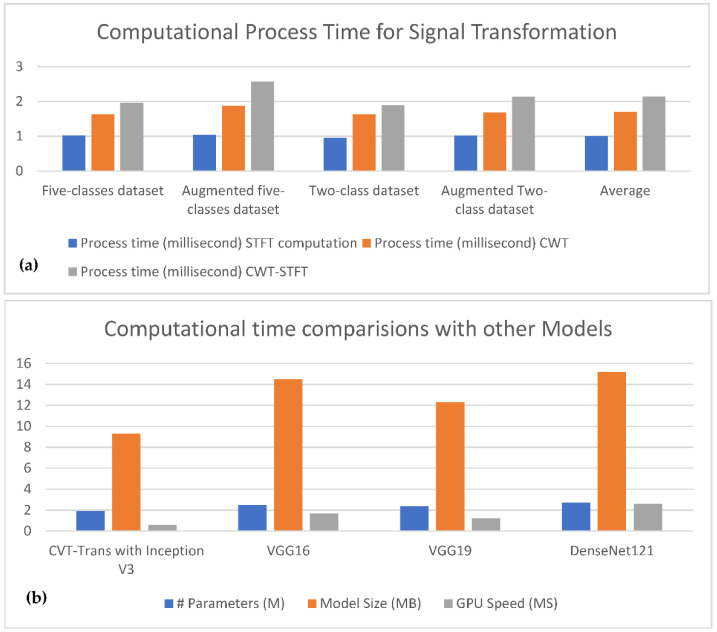
Computational comparisons with other transfer learning models. (**a**) The 1-D signal transferred into 2D CVT-STFT, (**b**) Classification computational cost.

**Figure 6 diagnostics-12-03109-f006:**
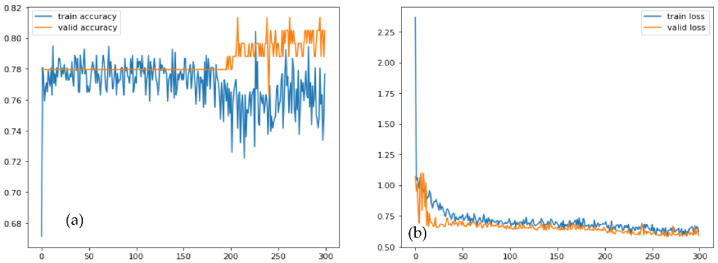
(**a**) Training and validation accuracy, (**b**) Training and validation loss graphs of the proposed architecture.

**Figure 7 diagnostics-12-03109-f007:**
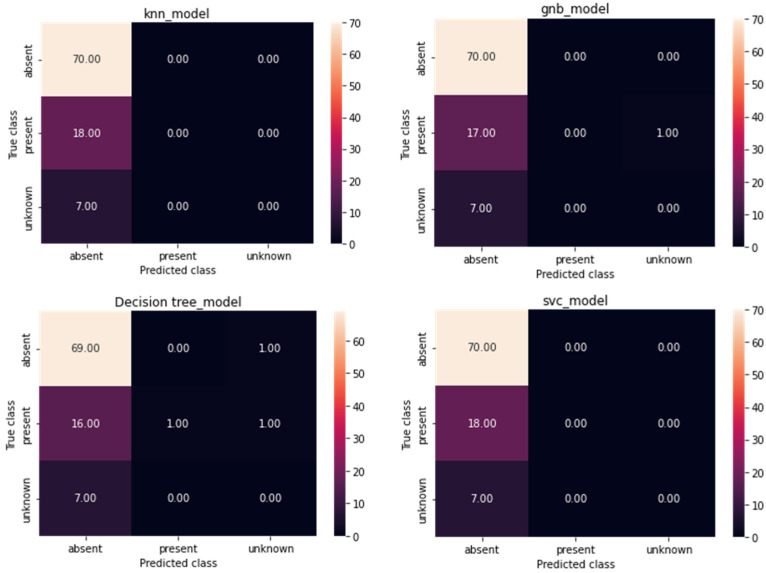
The confusion matrix for the proposed technique for classification of cardiovascular disease.

**Figure 8 diagnostics-12-03109-f008:**
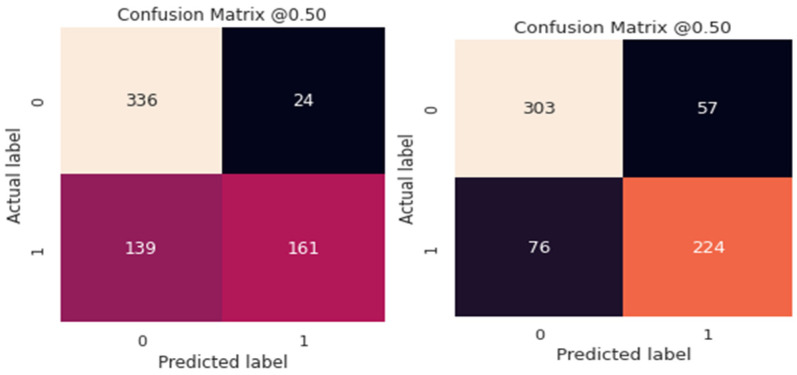
The confusion matrix for state-of-the-art techniques for classification of two categories of cardiovascular disease.

**Figure 9 diagnostics-12-03109-f009:**
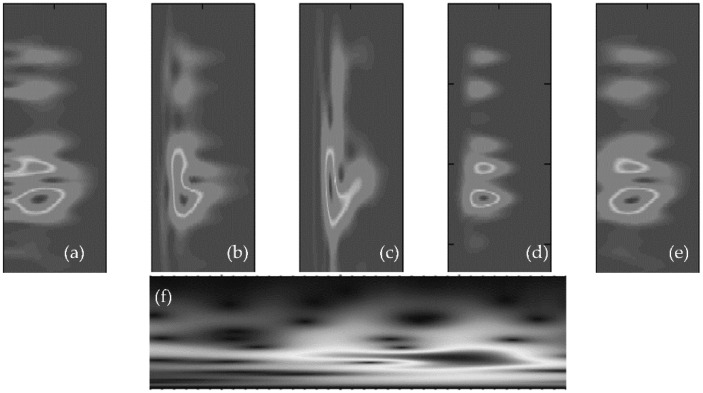
Time–frequency spectrum analysis of selected sample. (**a**) TF features, (**b**) STFT spectrogram and its inverse in (**c**,**d**), respectively, (**e**) CWT image, (**f**) Combination of CWT and STFT used in this paper.

**Figure 10 diagnostics-12-03109-f010:**
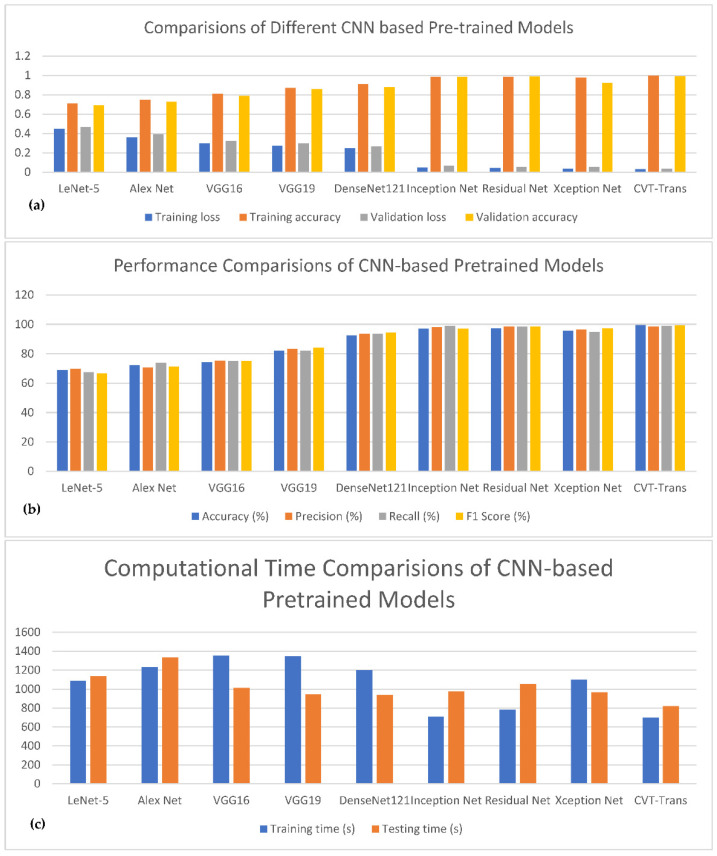
The comparisons of different CNN-based pretrained models with the proposed CVT-Trans technique for classification of five categories of cardiovascular disease. (**a**) Testing/Validation/Training loss, (**b**) Performance measures, (**c**) Computation time.

**Figure 11 diagnostics-12-03109-f011:**
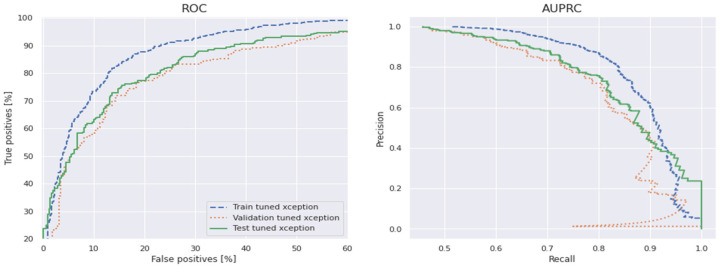
The ROC and AUPROC curves for proposed CVT-Trans technique for classification of five-categories cardiovascular disease.

**Table 2 diagnostics-12-03109-t002:** Classification results of the Proposed CVT-Trans system model based on 10-fold cross validation on selected dataset.

Predicted Classes	* SE	* SP	*F1-Score*	* RL	* PR	* ACC
NRM	100	0.96	0.97	97%	96%	99.4%
AS	0.98	0.95	0.95	95%	94%	99.4%
MVP	99.4	0.94	0.93	97%	95%	99.0%
MS	98.5	0.95	0.94	96%	94%	99.67%
MR	99.5	0.98	0.96	98%	98%	99.5%

* SE: Sensitivity, SP: Specificity, RL: Recall, PR: Precision, ACC: Accuracy, normal (NRM), aortic stenosis (AS), mitral valve prolapse (MVP), mitral stenosis (MS), and mitral regurgitation (MR).

**Table 3 diagnostics-12-03109-t003:** Classification results of the Proposed Model by using data augmentation, 10-fold cross validation test and by using five stages of cardiovascular disease.

10-Fold Cross Valid	* SE%	* SP%	*F1-Score*	* RL	* PR	* ACC
1	0.96	0.96	0.97	97%	96%	99.4%
2	0.92	0.92	0.95	95%	94%	98.4%
3	0.94	0.94	0.93	92%	95%	99.0%
4	0.95	0.95	0.94	93%	94%	98.4%
5	0.98	0.98	0.96	95%	98%	98.4%
6	0.97	0.97	0.95	94%	94%	98.0%
7	0.99	0.99	0.98	97%	98%	99.4%
8	0.98	0.98	0.98	96%	98%	99.0%
9	0.99	0.99	0.98	97%	97%	99.4%
10	0.99	0.98	0.99	98%	98%	100.0%

* SE: Sensitivity, SP: Specificity, RL: Recall, PR: Precision, ACC: Accuracy.

**Table 4 diagnostics-12-03109-t004:** Classification results of the Proposed Model by using data augmentation, 10-fold cross validation test and by using two stages of cardiovascular disease.

10-Fold Cross Valid	* SE	* SP	*F1-Score*	* RL	* PR	* ACC
1	0.95	0.96	0.97	97%	96%	99%
2	0.93	0.95	0.95	95%	94%	98%
3	0.94	0.94	0.96	92%	95%	99%
4	0.95	0.95	0.94	93%	94%	98%
5	0.98	0.98	0.96	95%	98%	98%
6	0.98	0.97	0.96	94%	94%	98%
7	0.99	0.98	0.98	97%	98%	99%
8	0.98	0.98	0.98	96%	98%	99%
9	0.99	0.99	0.98	97%	97%	99%
10	0.99	0.98	0.99	98%	98%	100%

* SE: Sensitivity, SP: Specificity, RL: Recall, PR: Precision, ACC: Accuracy.

**Table 5 diagnostics-12-03109-t005:** State-of-the-art classification results for categorizing five classes of PCG signals.

Studies	Features Extraction	Classifier	* ACC
Z.H. Wang et al. [17]	CWT + Spectrogram	LSTM-RNN	93%
A.M. Alqudah [18]	Instantaneous frequency-based features	RF and KNN	95%
CVT-Trans Model	CWT + Spectrogram	CVT + ATTF	99%

* CWT: Continuous wavelet transform, ACC: Accuracy, RF: Random Forest classifier, KNN: k-means clustering algorithm.

**Table 6 diagnostics-12-03109-t006:** Performance comparisons between proposed method and state-of-the-art techniques by using data augmentation, 10-fold cross validation test and by using two stages of CVD disease.

Methods	* SE	* SP	*F1-Score*	* RL	* PR	* ACC
Cheng-FRED [19]	0.75	0.74	0.75	76%	75%	75%
Rath- RF-MFO-XGB [20]	0.85	0.86	0.84	85%	84%	85%
Li-PCA-TSVM [21]	0.87	0.85	0.86	84%	86%	87%
Khan-ANN-LSTM [22]	0.88	0.87	0.88	85%	88%	86%
Saputra-NN-PSO [24]	0.78	0.76	0.75	77%	77%	77%
Arsalan-RF [25]	0.73	0.74	0.75	74%	74%	75%
**Proposed CVT-Trans**	0.99	0.98	0.99	98%	98%	100%

* SE: Sensitivity, SP: Specificity, RL: Recall, PR: Precision, ACC: Accuracy.

**Table 7 diagnostics-12-03109-t007:** Performance comparisons between proposed method and state-of-the-art techniques by using data augmentation, 10-fold cross validation test and by using five stages of CVD disease.

Methods	* SE	* SP	*F1-Score*	* RL	* PR	* ACC
Cheng-FRED [19]	0.70	0.72	0.71	70%	70%	71%
Rath- RF-MFO-XGB [20]	0.73	0.74	0.73	73%	73%	72%
Li-PCA-TSVM [21]	0.74	0.75	0.75	74%	74%	73%
Khan-ANN-LSTM [22]	0.78	0.79	0.77	78%	76%	77%
Saputra-NN-PSO [24]	0.77	0.78	0.76	77%	75%	75%
Arsalan-RF [25]	0.74	0.75	0.76	74%	75%	76%
**Proposed CVT-Trans**	0.99	0.98	0.99	98%	98%	100%

* SE: Sensitivity, SP: Specificity, RL: Recall, PR: Precision, ACC: Accuracy.

## Data Availability

The dataset of the five categories of original HS can be obtained in the repository: https://github.com/yaseen21khan/Classification-of-Heart-Sound-Signal-Using-Multiple-Features- (accessed on 1 January 2022).

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
