# Peer review of "Automatic Detection and Classification of Cardiovascular Disorders Using Phonocardiogram and Convolutional Vision Transformers"

_diagnostics, 2022, doi:10.3390/diagnostics12123109_

Round 1

Reviewer 1 Report

1.This manuscript is very well written and contributes to the (CVT-Trans) protocol to help patients diagnose heart valve problems. The new method has better indicators than the old one.

2. In Figure 4, (a) and (b) are not labeled.

3. the symbol spect is not explained in Formula (1) of 3.2.3.

4. In Section 4.2, why were only 10% of the 1800 original recordings of experimental data used

5. This paper is lack of sufficient explanation of the results. It is suggested to explain your results in detail and why you got such results.

Author Response

Reviewer 1:

Comment - (1) This manuscript is very well written and contributes to the (CVT-Trans) protocol to help patients diagnose heart valve problems. The new method has better indicators than the old one.

Response 1: Thank you reviewer to admire our manuscript. Yes, we have proposed a computational efficient solution compared to state-of-the-art systems for better diagnosis of heart disorder by using CVT-trans.  

Comment - (2) . In Figure 4, (a) and (b) are not labeled.

Response 2: As suggested by reviewer #1, we have remove more labels (a) and (b) more emphasis to show the labels in Figure 4.

Thank you to clear this problem in our paper.

Comment - (3) the symbol “spect” is not explained in Formula (1) of 3.2.3.

Response 3: Yes, you are right. We have added explanation of word “spect” in the section 3.2.3. You can check it here as follows.

Comment - (4) In Section 4.2, why were only 10% of the 1800 original recordings of experimental data used.

Response 4: Yes, you are right. We have changed these sentences as initially this was planed to used 10% but now, we have added all experimental results in terms of 10-fold cross validation results. These changes can be easily seen in the revised paper in the experimental section.

Thank you to highlight this issue.

Comment - (5) This paper is lack of sufficient explanation of the results. It is suggested to explain your results in detail and why you got such results.

Response 5: Yes, you are right. We have changed these sentences. All experimental results are re-written and added new explanation of all parts. You can find these changes in the experimental result analysis. We have added Computational analysis, new comparisons are added and new figures are also added. Those results indicate that the proposed model is outperformed compared to other approaches.

Thank you to highlight this issue.

Reviewer 2 Report

The manuscript entitle "Automatic Detection and Classification of Cardiovascular Disorders using Phonocardiogram and Convolutional Vision Transformer" presents a patch-embedding technique (CVT-Trans) based on a convolutional vision transformer to recognize the PCG signals into five classes. Here are my comments:

-The provided link to the dataset is invalid.

-What transfer learning models were used in this work? Justification should be presented.

-The detail of CWT used in this work must be further expanded.

-The authors should perform the experiment to validate the CWT as a TF with the other TFs to ensure the concluding remark as claim by the authors.

-Please provide the complexity of the proposed method.

Author Response

Reviewer #2:

Comment - (1) -  The provided link to the dataset is invalid.

Response 1: As suggested by reviewer #2, we have updated the dataset link. It was changed by the authors who provided the dataset. The link is updated as:

The dataset of the five categories of original HS can be obtained in the repository:https://github.com/yaseen21khan/Classification-of-Heart-Sound-Signal-Using-Multiple- Features-.

Thank you to clear this problem in our paper.

Comment - (2) What transfer learning models were used in this work? Justification should be presented.

Response 2: Yes, you are right. We have added the explanation of TL algorithm. We have used Inception v3 model by removing the FC layer as explained below. In addition, we have done new experiments to show the Inception v3 model compared to other TL models. On page#8, we have added following sentences to support it.

To develop CVT model, we employed Inception v3 as a pretrain model by removing last layer as it is used to ex-tract deep features. Compared to other pre-trained models such as VGG16 or VGG19, this paper is used the Inception v3 architecture to influence of computational efficiency and low parameters. The steps for classification are explained in the subsequent paragraphs.

Thank you to point out this viewpoint.

Comment - (3) The detail of CWT used in this work must be further expanded.

Response 3: Yes, you are right. We have added new sentences to explain further CWT as suggested by reviewer 2. Thank you, this valuable comment. We have revised the section 3.2.3 to explain further CWT signal transformation step.

We have also added new results to describe the importance of CWT and STFT transformation steps by describing further results. A new figure has been added in the paper to explain it.

Thank you to increase the quality of the paper.

Comment - (4) The authors should perform the experiment to validate the CWT as a TF with the other TFs to ensure the concluding remark as claim by the authors.

Response 4: Yes, you are right. As advised by you we have added new paragraphs in the result section and and also new diagram in the result section (figure 8) to show the importance of CWT compared to TW. A revised section is described here as:

To determine the time-frequency representation of a sound, time-frequency (TF) 298

transformation is a popular method in the classification of speech events. Using TF repre- 299

sentation, a one-dimensional (1D) signal is converted into a three-dimensional (3D) image 300

representation. Following that, the most likely sound source is identified using the attrib- 301

utes that were derived from the transformation. Based on their analysis in [32], the authors 302

draw the conclusion that the continuous wavelet transform-based spectrogram (CWTS) 303

presents the TF content of PCG signals in the clearest representation [36]. Several authors' 304

analyses show that the CWTS process and the signal are represented in the form of a spec- 305

trogram. The heart-sound signal's magnitude spectrogram is computed for each sample. 306

The transfer learning models are tested and trained using these spectrograms. We em- 307

ployed CWTS process compared to a technique [17] for recovering heart sound signals 308

based on LSTM architecture. 309

The "scale" parameter of the wavelet transform can be altered to identify various fre- 310

quencies in the signal as well as their locations. We now know the frequencies present in 311

the time signal as well as their locations. Wavelets are compressed at smaller scales. It can 312

therefore collect higher frequencies. A wider scale, on the other hand, can pick up lower 313

frequencies. An illustration of a compressed and stretched wavelet can be seen in the im- 314

age below. Superior time and frequency resolution is provided by the CWT. This enables 315

the use of various analysis windows of varying sizes and frequencies. The spectrograms 316

of the heart sound signals show the frequencies at different times and offer a visual rep- 317

resentation that can be utilized to distinguish individual heart sounds. The CWT produces 318

data for a spectrogram, and each RGB image is downsized to an array of size (n-by-m-by- 319

3) to match the inputs of various deep learning (DL) algorithms. Figure 4 displays the six 320

typical HS signal spectrograms.”

Thank you, this valuable comment.

Comment - (5) Please provide the complexity of the proposed method..

Response 5: Yes, you are right. We have performed further experiments to show the computational time and added new sub-section in the revised paper as:

4.4. Computational Analysis

To measure the performance of proposed CVT-Trans system, we have computed the  running time. To show that the suggested technique is a computational efficient model, it 606 was developed and trained on a GPU-based system rather than a CPU-based system. For 607 all datasets, the durations for STFT calculation, CVT with time domain input, and CVT- 608 Trans with frequency input were computed. On average, this step took 0.4 Ms. Overall, a 609 CWT spectrogram transform steps from original 1-D PCG signals on average took 1.2 MS. 610 This point-of-view is visually explained in Fig.5a. An attention-based CVT transformer 611 architecture is created in this study to categorize the spectrogram into five groups. On 612 average, this step took 1.2 MS. The key benefits of the proposed technique are quick clas- 613 sification and STFT computation, excellent accuracy acquired by utilizing all datasets, and a minimal number of layers. The original TL models contain more parameters and FLOPs 615 (given in Fig. 5b) compared to the suggested CVT-Trans model. 616 The outcome is a compact model that requires fewer computational support systems 617 to implement. Training time complexity = O(n*m*d), Where, the parameter n is the input 618 dimension, d is the batch size, and m is the output dimension. In general, the proposed 619 CVT-Trans model took linear running time is calculated as O(n*m*d). By utilizing tensor 620 processing units (TPUs), which are offered by the Google cloud, this time complexity can 621 be further decreased. In actual use, the TPUs significantly increased DL model speeds 622 while using less power. This viewpoint will also be covered in further studies.

A new figure also added to explain.

Thank you, for this valuable comment.

Reviewer 3 Report

Weakness

Method

1.     What is the difference between the proposed continuous wavelet transform-based spectrogram (CWTS) and CWT in [17]?

2.     What is the difference between the proposed convolutional vision transformer and [38]?

3.     I would like to know why the authors fuse the LBP and CNN features to form the feature map.

Experiment

1.     There is no need to times 100 when calculating the metrics.

2.     What dataset does the pre-training conduct on?

3.     Please provide the benchmark with the methods mentioned in Sec. 2 to compare the effectiveness of the proposed method.

4.     Which result can support the statement of “Overall classification accuracy for the experiment's five binary classification cases was 100%, 99.00%, 99.75%, 99.75%, and 99.60%, respectively.”?

Writing

1.     The authors should re-write the introduction of phonocardiography in Sec. 1, which is logically confusing.

2.     The authors should put more effort into describing the motivation of the proposed method in Sec. 1 instead of simply describing the contribution in bullet points. In other words, the description of PCG is redundant and should be shortened.

3.     Could the authors explain what “The categorization effect could be a subpart of deep networks with intricate and deep architecture” stands for? And what is the connection between the above sentence and “Thus, for the multiple classifications of cardiac disorders, deep features are required.”?

4.     The authors should re-draw Fig. 4 with clear legends or names to help readers to understand the connection between modules and the description in the paragraph.

5.     There has no Table 5 in the manuscript.

6.     Accuracy is falsely marked with “%”.

7.     Fig. 3 is confusing. Please correct the figure with five examples.

8.     In line 39, page 1, the reference to the figure is missing.

9.     In line 108, page 3, a period is missing.

10.   In line 473, page 10, a period is missing.

11.   In line 483, page 11, there is no Sec. 3.5.2 in the manuscript.

Author Response

Reviewer #3:

Comment - (1) -  What is the difference between the proposed continuous wavelet transform-based spectrogram (CWTS) and CWT in [17]?.

Response 1: As suggested by reviewer #3, we have updated the paragraph to show the CWT and STFT techniques to just use CWT in paper [17]].

Those changes can be easily seen in the signal transformation section.

We have done the following updated related to this comment.

To determine the time-frequency representation of a sound, time-frequency (TF) 298

transformation is a popular method in the classification of speech events. Using TF repre- 299

sentation, a one-dimensional (1D) signal is converted into a three-dimensional (3D) image 300

representation. Following that, the most likely sound source is identified using the attrib- 301

utes that were derived from the transformation. Based on their analysis in [32], the authors 302

draw the conclusion that the continuous wavelet transform-based spectrogram (CWTS) 303

presents the TF content of PCG signals in the clearest representation [36]. Several authors' 304

analyses show that the CWTS process and the signal are represented in the form of a spec- 305

trogram. The heart-sound signal's magnitude spectrogram is computed for each sample. 306

The transfer learning models are tested and trained using these spectrograms. We em- 307

ployed CWTS process compared to a technique [17] for recovering heart sound signals 308

based on LSTM architecture.

Thank you to clear this problem in our paper.

Comment - (2) -  What is the difference between the proposed convolutional vision transformer and [38]?

Response 2: As suggested by reviewer #3, we have updated the paragraphs.

Those changes can be easily seen in the signal transformation section.

We have done the following updated related to this comment.

The Vision Transformer (ViT) [37] is probably the first entirely transformer-based 417 design for vision, treating image patches as simple word sequences that are then encoded 418 using a transformer. ViT can produce impressive results in image recognition when it is 419 pretrained on huge datasets. ViT, however, has been found to perform poorly in image 420 (a) (b) (c) (d) (e) Diagnostics 2022, 12, x FOR PEER REVIEW 10 of 25 recognition without significant pre-training. This is a consequence of the Transformers' 421 strong model capability and lack of inductive bias, which causes overfitting. In multiple 422 subsequent studies, sparse Transformer models designed for visual tasks like local atten- 423 tion have been investigated to effectively regularize the model's capacity and enhance its 424 scalability. One such effective attempt to change transformers by applying self-attention 425 to shifted, non-overlapping windows is the Swin Transformer. For the first time, with a 426 pure vision transformer, this methodology outperformed ConvNets on the ImageNet 427 benchmark. Window-based attention has been found to have limited model capacity due 428 to the loss of non-locality and thus scales negatively on bigger data regimes like 429 ImageNet-21K, despite being more adaptable and generalizable than the complete atten- 430 tion utilized in ViT. However, full-attention acquisition of global interactions in a hierar- 431 chical network at early or high-resolution stages involves computationally intensive effort 432 since the attention operator has quadratic complexity. It is still challenging to include 433 global and local interactions to balance model capacity and generalizability within a com- 434 puting budget. 435 Shift, scale, and distortion invariance are characteristics of convolutional neural net- 436 works (CNNs) that have been transferred to the ViT architecture [38] while retaining the 437 benefits of Transformers (i.e., dynamic attention, global context, and better generaliza- 438 tion). Even though vision transformers are successful on a large scale, their performance 439 is still inferior to that of smaller CNN competitors (such as ResNets) when trained on less 440 input. One rationale might be that CNNs are better suited to addressing vision-related 441 issues because they naturally possess certain desirable qualities that ViT lacks. By utilizing 442 local receptive fields, shared weights, and spatial subsampling, a texture compels the cap- 443 ture of this local structure and, as a result, also achieves some degree of shift, scale, and 444 distortion invariance. Images, for instance, often contain a strong 2D local structure with 445 highly connected spatially neighboring pixels. Additionally, the hierarchical structure of 446 convolutional kernels enables the learning of a variety of complex visual patterns, from 447 low-level edges and textures to higher-order semantic patterns that incorporate local spatial context.

Comment - (3) -  I would like to know why the authors fuse the LBP and CNN features to form the feature map. Experiment?

Response 3: As suggested by reviewer #3, we have updated the paragraphs to show the importance of LBB and CNN features that used local and global interactions of pixels for attention layer.

Comment - (4) -  1.     There is no need to times 100 when calculating the metrics.

  1. What dataset does the pre-training conduct on?
  2. Please provide the benchmark with the methods mentioned in Sec. 2 to compare the effectiveness of the proposed method.
  3. Which result can support the statement of “Overall classification accuracy for the experiment's five binary classification cases was 100%, 99.00%, 99.75%, 99.75%, and 99.60%, respectively.”? Writing?

Response 4: As suggested by reviewer #3, we have updated the paragraphs to show the pre-training step and also the results part. All tables have been re-written and new experiments have been added to support the proposed model. A new result section has been re-written by writing 10-fold cross validation test. To perform pretraining, we used features set on selection dataset that is described in the data acquisition section.

Thank you to highlight this issue in the manuscript.

Comment - (5) The authors should re-write the introduction of phonocardiography in Sec. 1, which is logically confusing.? The authors should put more effort into describing the motivation of the proposed method in Sec. 1 instead of simply describing the contribution in bullet points. In other words, the description of PCG is redundant and should be shortened.

Response 5: As suggested by reviewer #3 and Reviewer#2, we have updated the paragraphs to show the importance of PCG signals used in this paper.

We have revised the introduction part as follows

The main cause of death worldwide is cardiovascular disease (CVD), which claims 29 more than 17 million lives each year. [1]. The CVD disease creates other pathological [2] 30 issues with the heart, heart valves, or blood vessels. Nowadays, the authors describe a 31 cost-effective and non-invasive technique for capturing heart signals through phonocar- 32 diography (PCG) [3,4]. It aids in enhancing the diagnosis of cardiac disorders and in cre- 33 ating new perceptions regarding the connection between the signal and the mechanical 34 function of the heart. PCG signals can be used to diagnose a variety of CVD signals, in- 35 cluding mitral stenosis (MS), mitral regurgitation (MR), aortic stenosis (AS), and mitral 36 valve prolapse (MVP). A visual example of these PCG signals categorized into five classes 37 as display in Fig 1. 38 In practice, the visual screening of the PCG signal takes time [5] and is error-prone. 39 Still, the arbitrary PCG signal inspection and analysis required by doctors require sub- 40 stantial training and expertise. This encourages the creation of a computer-aided diagnos- 41 tic (CAD) method for the recognition of PCG signal-based cardiac screening and abnor- 42 mality detection. The CVD classification is currently a promising topic of research based 43 on biomedical signal processing and artificial intelligence (AI) [6]. Techniques utilizing AI 44

can be utilized to get around these restrictions. Machine learning (ML) is a branch of AI 45 that entails feature selection, statistical analysis, salient feature extraction (SFA), and clas- 46 sification. ML techniques are extensively used in combination with PCG signals to detect 47 heart sounds [7]. Recently published papers for the diagnosis of cardiac illnesses used a 48 variety of research and methodologies that have been suggested [8–10]. Unfortunately, 49 the accuracy was not adequate, so the focus of the effort is on developing a very accurate 50 ML or DL for diagnosing of cardiac problems. In the past, the authors used a variety of 51 feature extraction techniques and classifiers. However, those feature selection and classi- 52 fication techniques are hand-crafted, which are frequently having iterative trial and error. 53 To resolve this issue, a deep learning (DL) techniques are developed. 54 Currently, the DL algorithms are still used in the primary approach for detecting 55 heart sounds because smart detection of PCG technology, which has not yet been widely 56 adopted in actual clinical diagnosis. Therefore, advancements in the field of CVD diagno- 57 sis is facilitated by the study of and deployment of computer-aided (CAD) heartbeat de- 58 tection techniques. According to the past, cardiovascular disease was mostly detected us- 59 ing four steps: (1) preprocessing of the HS signals, (2) feature extraction, (3) feature selec- 60 tion, and (4) identification of normal and abnormal HS recordings.

Thank you to highlight this issue in the manuscript.

Comment - (6) Could the authors explain what “The categorization effect could be a subpart of deep networks with intricate and deep architecture” stands for? And what is the connection between the above sentence and “Thus, for the multiple classifications of cardiac disorders, deep features are required.”?

Response 6: As suggested by reviewer #3, those sentences has been changed, which are not explained clear in the manuscript in the first version. In the revised version we have change sentences to show the importance of convolutional and vision transformer. Thank you to highlight this issue in the manuscript.

Comment - (7) The authors should re-draw Fig. 4 with clear legends or names to help readers to understand the connection between modules and the description in the paragraph.

Response 7: As suggested by reviewer #3, we have updated this figure. However, it is difficult to completely redraw in the first revision of the manuscript. As we have done major changes in the paper and it took almost 17 days to complete the revision. We have added new results and comparison in the first revision of the paper. If you will still insist, then we can do further update in the second revision. I hope you understand the situation.

Comment- (8) There has no Table 5 in the manuscript.

  1. Accuracy is falsely marked with “%”.
  2. Fig. 3 is confusing. Please correct the figure with five examples.
  3. In line 39, page 1, the reference to the figure is missing.
  4. In line 108, page 3, a period is missing.
  5. In line 473, page 10, a period is missing.
  6. In line 483, page 11, there is no Sec. 3.5.2 in the manuscript.

.

Response 8: As suggested by reviewer #3, we have updated all those sentences as advised by you. Those changes can be easily seen in the revision version. Thank you.

Reviewer 4 Report

The article content is Novel, however related study is inadequate.  More literature survey to be added from recent years and comparison of experiment model with earlier to be performed. Review the article for English literature and fine tune for scientific writing. 

Author Response

Reviewer #4:

Comment - (1) The article content is Novel, however related study is inadequate.  More literature survey to be added from recent years and comparison of experiment model with earlier to be performed. Review the article for English literature and fine tune for scientific writing.?.

Response 1: As suggested by reviewer #4, we have updated the related section and paper has been updated in terms of English writing.

Recent articles are added as suggested by you.

  1. Ghosh, S. K., Ponnalagu, R. N., Tripathy, R. K., Panda, G., & Pachori, R. B. (2022). Automated Heart Sound Activity Detection From PCG Signal Using Time–Frequency-Domain Deep Neural Network. IEEE Transactions on Instrumenta-tion and Measurement, 71, 1-10.
  2. Bao, X., Xu, Y., & Kamavuako, E. N. (2022). The Effect of Signal Duration on the Classification of Heart Sounds: A Deep Learning Approach. Sensors, 22(6), 2261.
  3. Ismail, S., Ismail, B., Siddiqi, I., & Akram, U. (2023). PCG classification through spectrogram using transfer learning. Biomedical Signal Processing and Control, 79, 104075.
  4. Tian, G., Lian, C., Zeng, Z., Xu, B., Su, Y., Zang, J., ... & Xue, C. (2022). Imbalanced Heart Sound Signal Classification Based on Two-Stage Trained DsaNet. Cognitive Computation, 1-14.
  5. Rezaee, K., Khosravi, M. R., Jabari, M., Hesari, S., Anari, M. S., & Aghaei, F. (2022). Graph convolutional network‐based deep feature learning for cardiovascular disease recognition from heart sound signals. International Journal of Intelligent Systems.
  6. Malik, H., Bashir, U., & Ahmad, A. (2022). Multi-classification neural network model for detection of abnormal heartbeat audio signals. Biomedical Engineering Advances, 4, 100048.

Those changes can be easily seen throughout the paper.

Round 2

Reviewer 2 Report

The authors have addressed my comments. I have no more concerns.

Author Response

Reviewer 1:

Comment - (1) (x) English language and style are fine/minor spell check required and Results can be improved.

Response 1: Thank you reviewer to for these important concerns. Yes, we have improved the English writing of the paper and we have done spell checker to correct the writing part.

In addition, results are improved as suggested by Reviewer#1 and Reviewer#2. We have done new comparison among CVT-Trans and state-of-the-art systems and added new Table 6 and Table 7. The results are improved. These changes can be easily seen in Page# 20 as:

We have also performed experiments to show the importance of the proposed CVT-Trans system compared to state-of-the-art approaches such as Cheng-FRED [19], Rath-RF-MFO-XGB [20], Li-PCA-TSVM [21], Khan-ANN-LSTM [22], Saputra-NN-PSO [24], and Arsalan-RF [25] in terms of SE, SP, F1-score, RL, PR, and ACC measures. The standard hyper-parameters are defined as presented in the corresponding studies. Firstly, those comparisons are performed between the proposed method and state-of-the-art techniques by using data augmentation and a 10-fold cross-validation test in terms of the two stages of CVD disease as shown in Table 6. As displayed in Table 6, the Khan-ANN-LSTM [22] system obtains good classification results (SE of 0.88, SP of 0.87, F1-score of 0.88, RL of 85%, PR of 88%, and ACC of 86%) for CVD heart disease compared to other approaches. However, the proposed CVT-Trans method achieved 100% ACC. Secondly, we have also measured the performance of the CVT-Trans system in terms of the recognition of the five stages of CVD heart disease. Those results are mentioned in Table 7. As mentioned in Table 7, the proposed CVT-Trans system outperformed (SE of 0.99, SP of 0.98, F1-score of 0.99, RL of 98%, PR of 98%, and ACC of 100%) compared to state-of-the-art approaches. Those good results are obtained because we have developed a convolutional vision transformer (CVT) architecture based on local and global attention mechanisms in a continuous wavelet transform-based spectrogram (CWTS) strategy. The developed strategy is effective and efficient compared to many state-of-the-art systems.

Methods

*SE

*SP

F1-score

*RL

*PR

*ACC

Cheng-FRED [19]

0.70

0.72

0.71

70%

70%

71%

Rath- RF-MFO-XGB [20]

0.73

0.74

0.73

73%

73%

72%

Li-PCA-TSVM [21]

0.74

0.75

0.75

74%

74%

73%

Khan-ANN-LSTM [22]

0.78

0.79

0.77

78%

76%

77%

Saputra-NN-PSO [24]

0.77

0.78

0.76

77%

75%

75%

Arsalan-RF [25]

0.74

0.75

0.76

74%

75%

76%

Proposed CVT-Trans

0.99

0.98

0.99

98%

98%

100%

Table 6 Performance comparisons between proposed method and state-of-the-art techniques by using data augmentation, 10-fold cross validation test and by using two stages of CVD disease.

 * SE: Sensitivity, SP: Specificity, RL: Recall, PR: Precision, ACC: Accuracy

Table 7 Performance comparisons between proposed method and state-of-the-art techniques by using data augmentation, 10-fold cross validation test and by using five stages of CVD disease.

Methods

*SE

*SP

F1-score

*RL

*PR

*ACC

Cheng-FRED [19]

0.70

0.72

0.71

70%

70%

71%

Rath- RF-MFO-XGB [20]

0.73

0.74

0.73

73%

73%

72%

Li-PCA-TSVM [21]

0.74

0.75

0.75

74%

74%

73%

Khan-ANN-LSTM [22]

0.78

0.79

0.77

78%

76%

77%

Saputra-NN-PSO [24]

0.77

0.78

0.76

77%

75%

75%

Arsalan-RF [25]

0.74

0.75

0.76

74%

75%

76%

Proposed CVT-Trans

0.99

0.98

0.99

98%

98%

100%

* SE: Sensitivity, SP: Specificity, RL: Recall, PR: Precision, ACC: Accuracy

Thank you to highlight this issue.

Reviewer 3 Report

Weakness

Method

1.     How come the results in Tab. 2 and Tab. 3 in the original manuscript are different from the results in the revised version?

Experiments

1.     I would like to know the performance comparison between the proposed method and [25], [24], [22], [21], [20], and [19].

Writing

1.     The Model Size of VGG19 in Fig. 5 is missing.

2.     Please label the axis in Fig. 9.

3.     It is not acceptable to concatenate time and sore in one bar in Fig. 9. Besides, this makes the difference of performance unobservable.

Author Response

Reviewer #2:

Comment - (1) -  1.     How come the results in Tab. 2 and Tab. 3 in the original manuscript are different from the results in the revised version?.

Response 1: Yes, you are right. In the original submission, there was a mistake because when we have transferred the manuscript in the template provided the journal, the old tables were presented. Those results were obtained by 80% and 20% split ratio. Now, we have used 10-fold cross validation test and selected the highest results obtained by the CVT-Trans system.

Thank you to clear this problem in our paper.

Comment - (2) I would like to know the performance comparison between the proposed method and [25], [24], [22], [21], [20], and [19].

Response 2: Yes, you are right as also highlighted by the reviwer#1. We have added the new experiments to show the comparisons with the proposed systems compared to state-of-the-art systems.

In addition, results are improved as suggested by Reviewer#1 and Reviewer#2. We have done new comparison among CVT-Trans and state-of-the-art systems and added new Table 6 and Table 7. The results are improved. These changes can be easily seen in Page# 20 as:

We have also performed experiments to show the importance of the proposed CVT-Trans system compared to state-of-the-art approaches such as Cheng-FRED [19], Rath-RF-MFO-XGB [20], Li-PCA-TSVM [21], Khan-ANN-LSTM [22], Saputra-NN-PSO [24], and Arsalan-RF [25] in terms of SE, SP, F1-score, RL, PR, and ACC measures. The standard hyper-parameters are defined as presented in the corresponding studies. Firstly, those comparisons are performed between the proposed method and state-of-the-art techniques by using data augmentation and a 10-fold cross-validation test in terms of the two stages of CVD disease as shown in Table 6. As displayed in Table 6, the Khan-ANN-LSTM [22] system obtains good classification results (SE of 0.88, SP of 0.87, F1-score of 0.88, RL of 85%, PR of 88%, and ACC of 86%) for CVD heart disease compared to other approaches. However, the proposed CVT-Trans method achieved 100% ACC. Secondly, we have also measured the performance of the CVT-Trans system in terms of the recognition of the five stages of CVD heart disease. Those results are mentioned in Table 7. As mentioned in Table 7, the proposed CVT-Trans system outperformed (SE of 0.99, SP of 0.98, F1-score of 0.99, RL of 98%, PR of 98%, and ACC of 100%) compared to state-of-the-art approaches. Those good results are obtained because we have developed a convolutional vision transformer (CVT) architecture based on local and global attention mechanisms in a continuous wavelet transform-based spectrogram (CWTS) strategy. The developed strategy is effective and efficient compared to many state-of-the-art systems.

Methods

*SE

*SP

F1-score

*RL

*PR

*ACC

Cheng-FRED [19]

0.70

0.72

0.71

70%

70%

71%

Rath- RF-MFO-XGB [20]

0.73

0.74

0.73

73%

73%

72%

Li-PCA-TSVM [21]

0.74

0.75

0.75

74%

74%

73%

Khan-ANN-LSTM [22]

0.78

0.79

0.77

78%

76%

77%

Saputra-NN-PSO [24]

0.77

0.78

0.76

77%

75%

75%

Arsalan-RF [25]

0.74

0.75

0.76

74%

75%

76%

Proposed CVT-Trans

0.99

0.98

0.99

98%

98%

100%

Table 6 Performance comparisons between proposed method and state-of-the-art techniques by using data augmentation, 10-fold cross validation test and by using two stages of CVD disease.

 * SE: Sensitivity, SP: Specificity, RL: Recall, PR: Precision, ACC: Accuracy

Table 7 Performance comparisons between proposed method and state-of-the-art techniques by using data augmentation, 10-fold cross validation test and by using five stages of CVD disease.

Methods

*SE

*SP

F1-score

*RL

*PR

*ACC

Cheng-FRED [19]

0.70

0.72

0.71

70%

70%

71%

Rath- RF-MFO-XGB [20]

0.73

0.74

0.73

73%

73%

72%

Li-PCA-TSVM [21]

0.74

0.75

0.75

74%

74%

73%

Khan-ANN-LSTM [22]

0.78

0.79

0.77

78%

76%

77%

Saputra-NN-PSO [24]

0.77

0.78

0.76

77%

75%

75%

Arsalan-RF [25]

0.74

0.75

0.76

74%

75%

76%

Proposed CVT-Trans

0.99

0.98

0.99

98%

98%

100%

* SE: Sensitivity, SP: Specificity, RL: Recall, PR: Precision, ACC: Accuracy

Thank you to highlight this issue.

Comment - (3)    The Model Size of VGG19 in Fig. 5 is missing.

Response 3: Yes, you are right. In the revised version we have added the model size of VGG19 TL algorithm. This figure has been updated and this change can be easily seen on Page# 14 and Figure 5.

Thank you for this valuable comment to improve the quality of the paper.

Comment - (4)    2.     Please label the axis in Fig. 9 and  It is not acceptable to concatenate time and sore in one bar in Fig. 9. Besides, this makes the difference of performance unobservable..

Response 4: Yes, you are right. In the revised version, we have revised the fig8.

Thank you for this valuable comment to improve the quality of the paper.

Figure 9. The comparisons of different CNN-based pretrained models with proposed CVT-Trans technique for classification of five-categories cardiovascular disease. Where, figure (a) shows the testing/validation/training loss and (b) represents the performance measures, and figure (c) shows the computation time.
